# Tanc2-mediated mTOR inhibition balances mTORC1/2 signaling in the developing mouse brain and human neurons

Sun-Gyun Kim [1,6], Suho Lee [1,6], Yangsik Kim [2], Jieun Park [3], Doyeon Woo[4], Dayeon Kim [2], Yan Li[1], Wangyong Shin [1], Hyunjeong Kang[3], Chaehyun Yook[3], Minji Lee[4], Kyungdeok Kim [1], Junyeop Daniel Roh [3], Jeseung Ryu [3], Hwajin Jung [1], Seung Min Um[3], Esther Yang[5], Hyun Kim[5], Jinju Han[2], Won Do Heo[3,4] & Eunjoon Kim[1,3 ✉]

mTOR signaling, involving mTORC1 and mTORC2 complexes, critically regulates neural development and is implicated in various brain disorders. However, we do not fully understand all of the upstream signaling components that can regulate mTOR signaling, especially in neurons. Here, we show a direct, regulated inhibition of mTOR by Tanc2, an adaptor/ scaffolding protein with strong neurodevelopmental and psychiatric implications. While *Tanc2*-null mice show embryonic lethality, *Tanc2*-haploinsufficient mice survive but display mTORC1/2 hyperactivity accompanying synaptic and behavioral deficits reversed by mTOR-inhibiting rapamycin. Tanc2 interacts with and inhibits mTOR, which is suppressed by mTOR-activating serum or ketamine, a fast-acting antidepressant. Tanc2 and Deptor, also known to inhibit mTORC1/2 minimally affecting neurodevelopment, distinctly inhibit mTOR in early- and late-stage neurons. Lastly, Tanc2 inhibits mTORC1/2 in human neural progenitor cells and neurons. In summary, our findings show that Tanc2 is a mTORC1/2 inhibitor affecting neurodevelopment.

[1] Center for Synaptic Brain Dysfunctions, Institute for Basic Science (IBS), Daejeon, Korea. [2] Graduate School of Medical Science and Engineering, Korea Advanced Institute for Science and Technology (KAIST), Daejeon, Korea. [3] Department of Biological Sciences, KAIST, Daejeon, Korea. [4] Center for Cognition and Sociality, Institute for Basic Science (IBS), Daejeon, Korea. [5] Department of Anatomy and Division of Brain Korea 21, Biomedical Science, College of Medicine, Korea University, Seoul, Korea. [6] These authors contributed equally: Sun-Gyun Kim, Suho Lee. ✉email: kime@kaist.ac.kr

Mammalian target of rapamycin (mTOR) signaling, a fundamental regulator of cellular growth and function[1–4], controls the development and function of the nervous system[5–12]. mTOR signaling is also strongly associated with various brain disorders, including brain tumors, epilepsy, neurodegenerative disorders (e.g., Alzheimer's and Parkinson's diseases), neurocutaneous diseases (e.g., tuberous sclerosis complex and neurofibromatosis), intellectual disability, autism spectrum disorders (ASD), depression, and diseases of abuse (e.g., alcoholism)[5–12].

mTOR nucleates the formation of mTOR complex-1 (mTORC1) and -2 (mTORC2) by associating with both shared and distinct components. These include mLST8 and Deptor (for both mTORC1 and mTORC2), Raptor and PRAS40 (mTORC1 only), and Rictor, Protor, and Sin1 (mTORC2 only)[1–4]. These mTOR-interacting proteins coordinate the activity, subcellular localization, and substrate interactions of mTOR[1–4]. Loss of proteins that enable or stimulate mTORC1/2 function, namely mTOR, Raptor, Rictor, and mLST, in mice leads to embryonic lethality[1]. However, deletion of Deptor or PRAS40 in mice has no significant impact on embryonic development or postnatal growth or survival[13,14]. There are upstream negative regulators of mTOR such as TSC1/2 and PTEN, NF1, and DEPDC5 that have strong impacts on neurodevelopment[5–12,15–20], although these regulators are not fully understood.

Tanc2, a large (~200 kDa) multi-domain adaptor/scaffolding protein, is highly expressed in the brain[21,22] and modestly expressed in other tissues (www.ebi.ac.uk/gxa/home)[23]. Tanc2 is present at both synaptic and non-synaptic sites in neurons. Synaptic Tanc2 directly interacts with PSD-95[21,22], an abundant excitatory postsynaptic scaffolding protein[24–26], and also promotes synaptic capture of motor protein-transported vesicles[27]. However, how Tanc2 regulates synaptic and neuronal functions is not fully understood.

The homozygous deletion of Tanc2 in mice leads to embryonic lethality[22]. In humans, TANC2 mutations are extensively associated with various neuropsychiatric disorders, including intellectual disability, ASD, developmental delays, and schizophrenia[23,28–36]. Disruptive TANC2 mutations were recently identified in 20 different patients with neurodevelopmental symptoms associated with psychiatric disorders[23]. These results suggest that Tanc2 is a critical regulator of brain development and function, but the underlying mechanisms remain unclear.

In this work, we show that Tanc2 interacts with and inhibits mTOR, and that Tanc2 deletion in mice leads to mTOR hyperactivity and synaptic and behavioral abnormalities that are responsive to the mTOR inhibitor rapamycin. Tanc2 and Deptor, a known inhibitor of mTOR, act at early and late postnatal stages, respectively. TANC2 knockdown in human neurons leads to mTOR hyperactivity. These results suggest that Tanc2 is a negative regulator of mTOR with neurodevelopmental impacts.

## Results

**Abnormal behaviors and synaptic plasticity in Tanc2-mutant mice.** To explore in vivo functions of Tanc2, we first characterized mice carrying a heterozygous deletion of the Tanc2 gene (Tanc2+/−), encoding the Tanc2 protein. Tanc2+/− mice, unlike homozygous Tanc2-mutant (Tanc2−/−) mice that show embryonic lethality[22], showed better survival. However, Tanc2+/− mice showed decreased postnatal survival rates (~56% at postnatal day 5 [P5] and ~44% at P110, relative to 100% expected values), indicative of substantial early (embryonic or early postnatal) lethality that is followed by moderate lethality during adolescence and adulthood. Body weights of Tanc2+/− mice were ~90% of those in wild-type (WT) mice. These results indicate a dose-

dependent impact of Tanc2 deletion on mouse development and survival.

In behavioral tests, adult male Tanc2+/− mice (2–5 months; male) showed impaired spatial learning and memory in the Morris water maze, but normal novel-object recognition (Fig. 1a and Supplementary Fig. 1a). These mice also displayed hyperactivity (open-field) and anxiolytic-like behavior (elevated plus-maze), but largely normal social behavior (three-chamber) and moderate anti-depression-like behavior (forced-swim but not tail-suspension), and as neonates, showed suppressed ultrasonic vocalizations upon mother separation (Supplementary Figs. 1b–e and 2). Female adult Tanc2+/− mice showed largely similar behavioral abnormalities; hyperactivity (open-field) and anxiolytic-like behavior (elevated plus-maze) but normal depression-like behavior (forced-swim and tail-suspension) (Supplementary Fig. 3). These results indicate that Tanc2+/− mice are more relevant to human disease conditions compared with Tanc2−/− mice.

To better understand the impaired spatial learning and memory in Tanc2+/− mice, we examined synaptic plasticity in the hippocampus. Long-term potentiation (LTP) induced by high-frequency stimulation (HFS) was suppressed at Schaffer collateral-CA1 pyramidal cell (SC-CA1) synapses of Tanc2+/− mice (7–8 weeks) (Fig. 1b). Input–output relationship and paired-pulse facilitation were unaltered at these synapses (7–8 weeks) (Fig. 1c, d), suggesting that basal excitatory synaptic transmission and presynaptic release are normal, respectively. In contrast, HFS-LTP at a younger age (4–5 weeks) was normal (Fig. 1e), suggestive of age-dependent LTP impairment. Input–output relationship and paired-pulse facilitation were also normal at these synapses (4–5 weeks) (Fig. 1f, g). Long-term depression (LTD) induced by low-frequency stimulation (LFS) was suppressed at Tanc2+/− SC-CA1 synapses (2–3 weeks), whereas mGluR-dependent LTD was normal (3–4 weeks) (Fig. 1h, i).

The abovementioned decrease in LFS-LTD at 2–3 weeks, which contrasts with the normal LTP at a similar age (4–5 weeks), cannot be explained by the decrease in currents of NMDA receptors (NMDARs), which are known to regulate both LTP and LTD[37,38]. We thus tested whether synaptic signaling downstream of NMDAR activation, also known to control LTP/LTD[37,38], is altered by immunoblot analysis of neuronal signaling proteins.

**Early postnatal mTOR hyperactivity in Tanc2-mutant mice.** Intriguingly, mTOR activity, indirectly measured by total levels of mTOR phosphorylation (S2448) in immunoblot analyses, was markedly (approximately fivefold) increased in the whole brain of Tanc2+/− pups (P14) without a change in total mTOR levels (Fig. 2a). This change was accompanied by hyper-phosphorylation of 4E-BP (T37/46), a downstream target of mTOR[1–4], but not S6 (S235/236), another mTOR target[1–4], likely owing to compensatory changes occurring in heterozygous mice (see the stronger changes induced by homozygous Tanc2 deletion, below). In contrast, activities of PI3K (phosphoinositide 3-kinase), PTEN (phosphatase and tensin homolog), and TSC1/2 (tuberous sclerosis 1/2)—signaling proteins upstream of mTOR—were normal (Fig. 2b), suggesting that they do not contribute to the mTOR hyperactivity.

Phosphorylation of Akt (S473), reflecting mTORC2 activity[1–4], was also strongly increased (Fig. 2a), suggesting that both mTORC1 and mTORC2 are hyperactive in the Tanc2+/− brain (P14). Moreover, Ser-9 phosphorylation of GSK3β (glycogen synthase kinase 3β), a downstream target of Akt[39] that promotes LTD[38], was increased (indicating reduced activity), in line with the suppressed LTD at Tanc2+/− hippocampal synapses. Phosphorylation of PKCα (S657), another mTORC2 substrate, however, was not altered in the mutant brain.

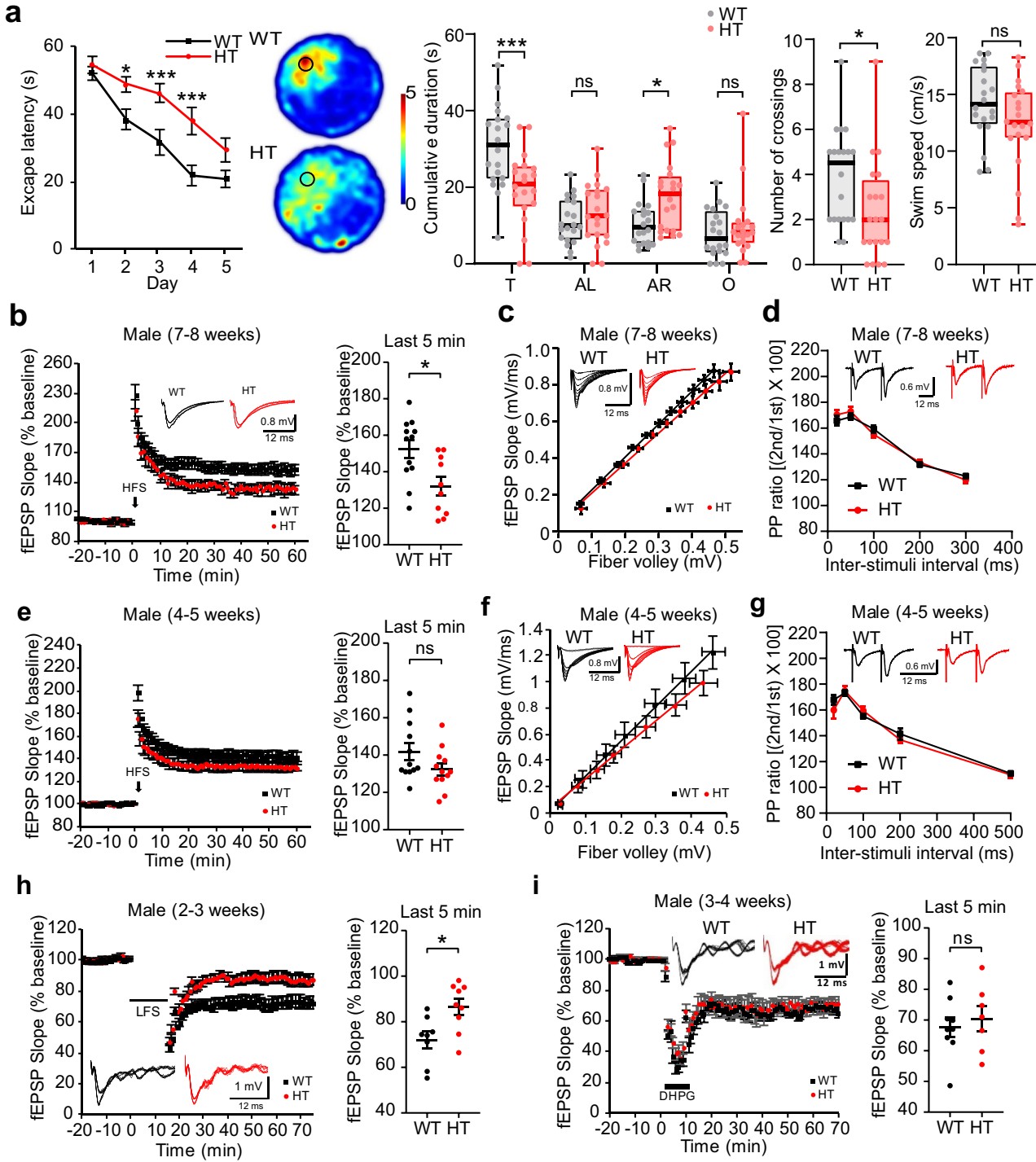

Interestingly, tests of *Tanc2*[+/−] juveniles (P28) and adults (P52) showed no significant changes in mTORC1 or mTORC2 activity, as indicated by immunoblot analyses of mTOR (S2448), S6 (S235/236), 4E-BP (T37/46), Akt (S473), and GSK3β (S9) (Fig. 2c, d). These results at P28 and P52 contrast with those from younger *Tanc2*[+/−] mice (P14) (Fig. 2a, b), which suggest that Tanc2 functions are age-dependent, consistent with the decreasing Tanc2 expression in the WT mouse brain after P14[22].

Because the mTOR hyperactivity observed in *Tanc2*[+/−] pups (P14) might represent indirect changes attributable to long-term deletion of *Tanc2*, we generated another *Tanc2*-mutant mouse line that carries a floxed *Tanc2* allele (*Tanc2*[fl/fl]) for use in creating a conditional gene knockout (cKO) (Supplementary

Fig. 4). Local homozygous knockout of *Tanc2* in the hippocampus of *Tanc2*[fl/fl] mice by the injection of AAV1-hSyn-Cre-EGFP at P5 and analysis at P14, but not the injection at P19 and analysis at P28, induced hyper-phosphorylation of S6 (S235/236), 4E-BP (T37/46), Akt (S473), GSK3β (S9), and mTOR (S2248) (Fig. 2e–h), indicative of mTORC1 and mTORC2 hyperactivity. These results collectively suggest that *Tanc2* deletion leads to mTORC1/2 hyperactivity at the pup (P7–14), but not juvenile (P21–28) or adult (~P52), stage.

**Early rapamycin treatment normalizes LTP and behaviors in adult *Tanc2*[+/−] mice.** To determine whether the early mTOR

**Fig. 1 Abnormal behaviors and synaptic plasticity in *Tanc2*-mutant mice. a** Impaired spatial learning and memory in *Tanc2*[+/−] mice (2–5 months; male) in the Morris water-maze test, as shown by the escape latency in the learning phase, the time spent in the target quadrant in the probe phase (T target; AR adjacent right; AL adjacent light; O opposite), and the number of exact crossings over the former hidden platform in the probe phase. Note that swim speed was normal in *Tanc2*[+/−] mice. Data values in line graphs represent mean ± SEM (standard error of mean), and those in box-and-whisker plots indicate median (mid-line), interquartile range (25–75%; box), and whole range (whisker). ($n = 20$ mice [WT] and [HT], $*P < 0.05$, $***P < 0.001$, ns not significant, two-way repeated measures/RM ANOVA and Student's $t$ test [number of crossings and swim speed]). **b** Suppressed HFS-LTP at SC-CA1 synapses of *Tanc2*[+/−] mice (7–8 week-old male). Data: mean ± SEM. ($n = 12$ slices, 5 mice [WT], and 10, 5 [HT], $*P < 0.05$, Student's $t$ test). **c** Normal basal transmission at SC-CA1 synapses of *Tanc2*[+/−] mice (7–8 weeks; male), as shown by initial fEPSP (field excitatory postsynaptic potential) slopes plotted against stimulus intensities (input–output). Data: mean ± SEM. ($n = 10$, 3 [WT] and [HT], two-way RM ANOVA [main genotype effect $P = 0.8051$]). **d** Normal paired-pulse facilitation at SC-CA1 synapses of *Tanc2*[+/−] mice (7–8 weeks; male), as shown by paired-pulse ratios plotted against inter-pulse intervals. Data: mean ± SEM. ($n = 10$, 3 [WT] and [HT], two-way RM ANOVA [main genotype effect $P = 0.9027$]). **e** Normal HFS-LTP at SC-CA1 synapses of 4–5-week-old *Tanc2*[+/−] mice (male). Data: mean ± SEM. ($n = 12$, 6 [WT], and 12, 5 [HT], ns not significant, Student's $t$ test). **f** Normal basal transmission at SC-CA1 synapses of *Tanc2*[+/−] mice (3–4 weeks; male), as shown by input–output curve. Data: mean ± SEM. ($n = 11$, 4 [WT], and 13, 5 [HT], two-way RM ANOVA [main genotype effect $P = 0.3725$]). **g** Normal paired-pulse facilitation at SC-CA1 synapses of *Tanc2*[+/−] mice (3–4 weeks; male). Data: mean ± SEM. ($n = 11$, 4 [WT], and 13, 5 [HT], two-way RM ANOVA [main genotype effect $P = 0.6515$]). **h** Suppressed LFS-LTD at SC-CA1 synapses of *Tanc2*[+/−] mice (2–3 weeks; male). Data: mean ± SEM. ($n = 8$, 6 [WT], and 9, 5 [HT], $*P < 0.05$, Student's $t$ test). **i** Normal mGluR-LTD induced by the group I mGluR agonist DHPG (50 μM, 10 min) at SC-CA1 synapses of *Tanc2*[+/−] mice (3–4 weeks; male). Data: mean ± SEM. ($n = 9$, 5 [WT], and 7, 4 [HT], ns not significant, Student's $t$ test). See Source Data 1 for raw data values and Supplementary Table 1 for statistical details.

hyperactivity in *Tanc2*[+/−] juveniles (P14) leads to the synaptic and behavioral abnormalities observed in *Tanc2*[+/−] adults (>P56), we chronically treated *Tanc2*[+/−] pups with the mTOR inhibitor rapamycin at early stages (5 mg/kg; intraperitoneal [i. p.]; 3 days/week; P10–35) and monitored synaptic and behavioral phenotypes in adult mice (Fig. 3a).

This treatment normalized the suppressed LTP in adult *Tanc2*[+/−] mice (>P56), without affecting WT synapses (Fig. 3b). In addition, it rescued the abnormal behaviors (spatial memory, hyperactivity, and anxiolytic-like behavior) in *Tanc2*[+/−] adults without affecting those in WT mice (Fig. 3c–e). These results suggest that early mTOR hyperactivity leads to late synaptic and behavioral abnormalities and that early correction of mTOR hyperactivity normalizes the mutant phenotypes in adults, highlighting long-lasting effects.

**Tanc2 interacts with and inhibits mTOR.** To gain mechanistic insight into how *Tanc2* deletion induces mTOR hyperactivity, we first tested whether Tanc2 interacts with mTOR using protein–protein binding assays. Purified Tanc2 protein interacted with purified mTOR protein (Fig. 4a). Tanc2 also formed a complex with mTOR in the mouse brain (Fig. 4b, c). This interaction was mediated by multiple regions of Tanc2 protein and the C-terminal region of mTOR containing FRB and kinase domains (Fig. 4d–f). Here, mTOR was found to additionally interact with Tanc1 (Fig. 4c), a relative of Tanc2 that is strongly expressed in late stages of rat brain development (>P14) and regulates synapse development but is not critical for mouse development[21,22]. In a control experiment, Tanc2 did not interact with Deptor (Fig. 4d), a known inhibitor of mTOR[40].

The results described thus far suggest that Tanc2 interacts with mTOR, but do not speak to whether Tanc2 inhibits the kinase activity of mTOR. We tested this possibility by overexpressing Tanc2 in HEK293T cells and found that this was sufficient to inhibit endogenous mTOR activity (Fig. 5a). Consistent with this, in vitro assays using purified proteins showed that Tanc2 inhibits mTOR kinase activity, as evidenced by decreased phosphorylation of the mTORC1 (mTOR + Raptor) target S6K in the presence of Tanc2 (Fig. 5b). In addition, Tanc2 decreased phosphorylation of the mTORC2 (mTOR + Rictor) target Akt (Fig. 5c).

**Serum and ketamine regulate the Tanc2–mTOR interaction.** We next investigated whether Tanc2–mTOR interactions are regulated by extracellular influences, first testing serum, which is known to activate mTOR[2]. Serum starvation promoted the colocalization and biochemical association of Tanc2 with mTOR in HEK293T cells within ~4 h. This effect was reversed by serum replenishment for ~24 h (Fig. 6a, b), suggesting that mTOR dissociates from Tanc2 upon serum stimulation. Moreover, the Tanc2–mTOR interaction induced by serum starvation was inhibited by rapamycin (Fig. 6c, d), suggesting that Tanc2 and rapamycin compete for binding to the mTOR FRB domain. Tanc1, which also associates with mTOR in the brain, interacted with mTOR in a serum- and rapamycin-dependent manner (Supplementary Fig. 5).

We next tested whether the Tanc1/2–mTOR interaction could be regulated in the brain of mice (P14) by ketamine-induced mTOR activation. Treatment with ketamine (10 mg/kg; i.p.), a fast-acting antidepressant known to stimulate mTOR signaling[41], rapidly (~30–60 min) increased mTOR activity and promoted synaptic localization of mTOR-associated proteins as well as PSD-95 (Supplementary Fig. 6), as previously reported[41]. Importantly, ketamine treatment suppressed the Tanc1/2–mTOR interaction without affecting the Tanc1/2–PSD-95 interaction (Fig. 6e)[21,22], suggesting that Tanc1/2 bridges mTOR to PSD-95 at the synapse in a regulated manner.

**Tanc2, Deptor, and Tanc1 distinctly inhibit mTORC1/2 in early- and late-stage neurons.** Because Deptor, similar to Tanc2, also binds and inhibits mTORC1/2[40], we tested whether Tanc2 and Deptor show overlapping or distinct spatiotemporal expression patterns. Immunoblot analyses using cultured neurons and mouse brain extracts showed that Tanc2 protein was more strongly expressed in early stages (embryonic and early postnatal) and was less enriched at synapses (Supplementary Fig. 7). In contrast, Deptor and Tanc1 showed progressive increases in expression across postnatal stages and stronger synaptic enrichment in both cultured neurons and mouse brains, a pattern similar to that reported for rat Tanc1 and Tanc2[21,22].

These results suggest that Tanc2 and Deptor/Tanc1 may distinctly inhibit mTOR activity at different developmental stages. We thus sought to acutely knockdown Tanc2 and Deptor/Tanc1 in cultured mouse hippocampal neurons during early (days in vitro [DIV] 7–14) and late (DIV 21–28) stages. Early-stage Tanc2 knockdown induced hyper-phosphorylation of S6 (S235/236), 4E-BP (T37/46), Akt (S473), and GSK3β (S9) (Fig. 7a, b), suggestive of mTORC1 and mTORC2 hyperactivity. Akt (T308) phosphorylation was unaltered, in line with the normal activities of mTOR-upstream proteins in *Tanc2*[+/−] mice (Fig. 2b).

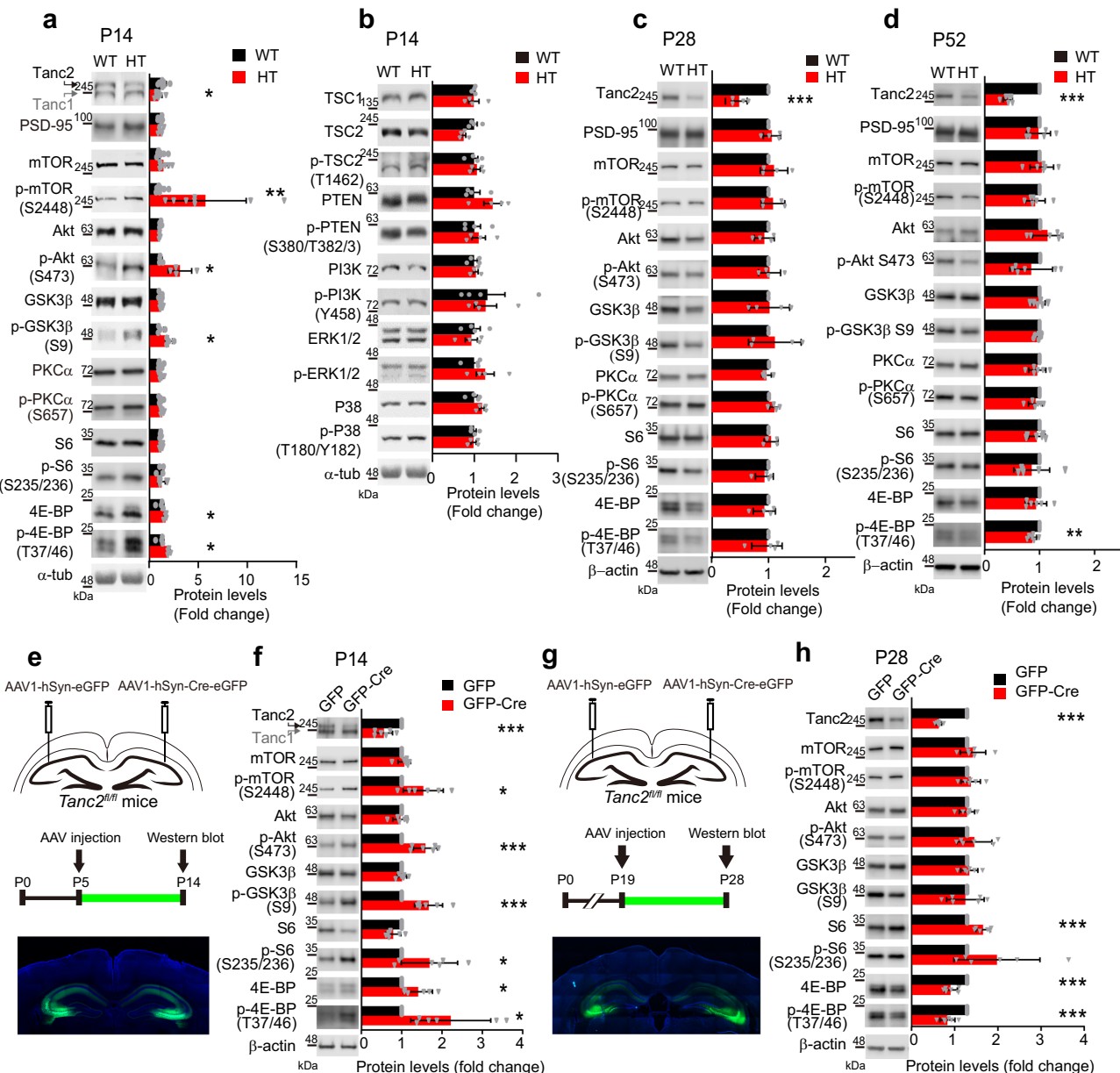

**Fig. 2 Increased mTOR activity in *Tanc2*-mutant mice. a** Increased mTOR activity (mTORC1 and mTORC2) in the brain of *Tanc2*$^{+/−}$ (HT) mice (P14; male), as shown by Ser-2448 phosphorylation. Note that phosphorylation levels of 4E-BP1 (Thr-37/46), but not S6 (Ser-235/236), downstream of mTOR were also increased. Note also that the increased phosphorylation of Akt (S473) indicates mTOR activation in the mTORC2 complex. For quantification, HT and WT signals were normalized to those of α-tubulin. See Source Data 2 for uncropped immunoblot images for this panel and all other immunoblot panels. Data: mean ± SD (standard deviation). Numbers on the left side of the immunoblots indicate molecular masses in kDa. ($n = 4$–8 mice [WT, HT], except for 8 for mTOR, p-mTOR, and Tanc1/2, *$P < 0.05$, **$P < 0.01$, ***$P < 0.001$, Student's $t$ test). **b** Lack of changes in phosphorylated or total levels of PI3K, PTEN, and TSC2 proteins upstream of mTOR in the *Tanc2*$^{+/−}$ brain (P14; male). Immunoblot signals were normalized to α-tubulin signals. Data: mean ± SD. ($n = 4$ mice [WT, HT], Student's $t$ test). **c** Normal phosphorylated and total levels of mTOR (S2448), S6 (S235/236), 4E-BP (T37/46), Akt (S473), and GSK3β (S9) in the *Tanc2*$^{+/−}$ brain (P28; male). Immunoblot signals were normalized to α-tubulin signals. Data: mean ± SD. ($n = 4$ animals [WT, HT], ***$P < 0.001$, Student's $t$ test). **d** Normal phosphorylated and total levels of mTOR (S2448), S6 (S235/236), 4E-BP (T37/46), Akt (S473), and GSK3β (S9) in the *Tanc2*$^{+/−}$ brain (P52; male). Immunoblot signals were normalized to α-tubulin signals. Note that 4E-BP phosphorylation is moderately decreased. Data: mean ± SD. ($n = 6$ animals [WT, HT], **$P < 0.01$, ***$P < 0.001$, Student's $t$ test). **e, f** Cre-dependent acute hippocampal *Tanc2* deletion during P5–14 leads to increased phosphorylation of S6 (S235/236), 4E-BP (T37/46), Akt (S473), GSK3β (S9), and mTOR (S2448), indicative of mTORC1 and mTORC2 hyperactivity. AAV1-Synapsin-Cre-eGFP and AAV1-Synapsin-eGFP (control) were injected in parallel into both sides of the hippocampus (CA3 region) of *Tanc2*$^{fl/fl}$ mice (eGFP expression is indicated in green). Note that S6 phosphorylation was increased, in contrast to the normal S6 phosphorylation observed in the *Tanc2*$^{+/−}$ brain (Fig. 2a). Data: mean ± SD. ($n = 6$ independent experiments, *$P < 0.05$, ***$P < 0.001$, Student's $t$ test). **g, h** Cre-dependent acute hippocampal (CA3 region) *Tanc2* deletion during P19–28 does not increase phosphorylation of S6 (S235/236), 4E-BP (T37/46), Akt (S473), GSK3β (S9), or mTOR (S2448), indicative of the lack of mTORC1 and mTORC2 hyperactivity. Note that total (not phosphorylation) levels of S6 were increased, and that total and phosphorylation levels of 4E-BP were decreased (not increased). Data: mean ± SD. ($n = 6$ independent experiments, ***$P < 0.001$, Student's $t$ test). See Source Data 1 for raw data values and Supplementary Table 1 for statistical details.

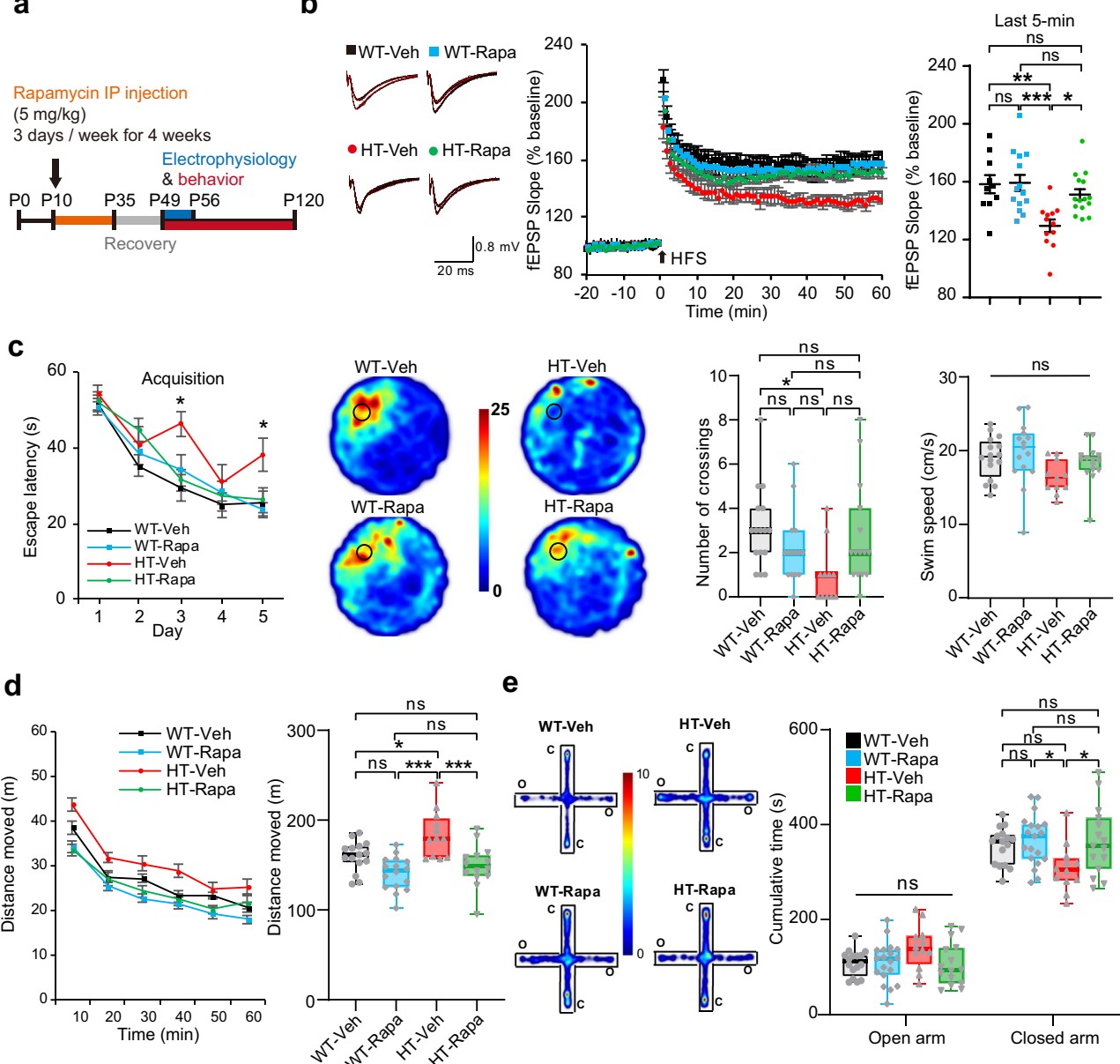

**Fig. 3 Early chronic rapamycin treatment normalizes LTP and abnormal behaviors in *Tanc2*+/− mice. a** A diagram showing early, chronic rapamycin treatment followed by electrophysiology and behavioral tests. **b** Early, chronic rapamycin treatment (P10–35) improves HFS-LTP at SC-CA1 synapses in adult *Tanc2*+/− mice (7–8 weeks). Data: mean ± SEM. (*n* = 10 slices, 5 mice [WT-Veh/vehicle], 14, 6 [WT-Rapa/rapamycin], 12, 5 [HT-Veh], and 15, 7 [HT-Rapa], *P < 0.05, **P < 0.01, ***P < 0.001, ns not significant, two-way ANOVA with Bonferroni test). **c** Early, chronic rapamycin treatment improves impaired spatial learning and memory in *Tanc2*+/− mice (2–4 months) in the Morris water-maze test, as indicated by escape latency and the number of crossings over the former platform location in the probe test. Data: mean ± SEM (line graphs), minimal, maximal, median, 25 and 75% values (box-whisker plots). (*n* = 16 animals [WT-Veh], 15 [WT-Rapa], 11 [HT-Veh], and 15 [HT-Rapa], *P < 0.05, ns not significant, two-way repeated measures/RM ANOVA with Bonferroni test [for escape latency] and one-way ANOVA with Bonferroni test [for number of crossings and swim speed]). **d** Early, chronic rapamycin treatment (P10–35) improves hyperactivity in *Tanc2*+/− mice (2–4 months) in the open-field test, as shown by distance moved over 60 min and total distance moved. Data: mean ± SEM (line graphs), minimal, maximal, median, 25 and 75% values (box-whisker plots). (*n* = 15 mice [WT-Veh/vehicle], 16 [WT-Rapa/rapamycin], 12 [HT-Veh], and 14 [HT-Rapa], *P < 0.05, ***P < 0.001, ns not significant, two-way ANOVA with Bonferroni test). **e** Early, chronic rapamycin treatment (P10–35) improves anxiolytic-like behaviors in *Tanc2*+/− mice (2–4 months) in the elevated plus-maze test, as shown by time spent in open/close arms. Data: minimal, maximal, median, 25 and 75% values. (*n* = 16 mice [WT-Veh], 19 [WT-Rapa], 12 [HT-Veh], and 15 [HT-Rapa], *P < 0.05, ns not significant, one-way ANOVA with Tukey test [Bonferroni test did not yield a significant difference between HT-Veh and HT-Rapa.]). See Source Data 1 for raw data values and Supplementary Table 1 for statistical details.

In contrast, late-stage Tanc2 knockdown (DIV 21–28) did not affect these phosphorylation events (Fig. 7c, d).

Late-stage, but not early-stage, knockdown of Deptor induced hyper-phosphorylation of S6 (S235/236), 4E-BP (T37/46), Akt (S473), and GSK3β (S9) (Fig. 7a–d), suggestive of mTORC1 and mTORC2 hyperactivity. In addition, late-stage knockdown of Tanc1 induced similar hyper-phosphorylation of S6 (S235/236), 4E-BP (T37/46), Akt (S473), and GSK3β (S9) (Fig. 7 c, d). Tanc2

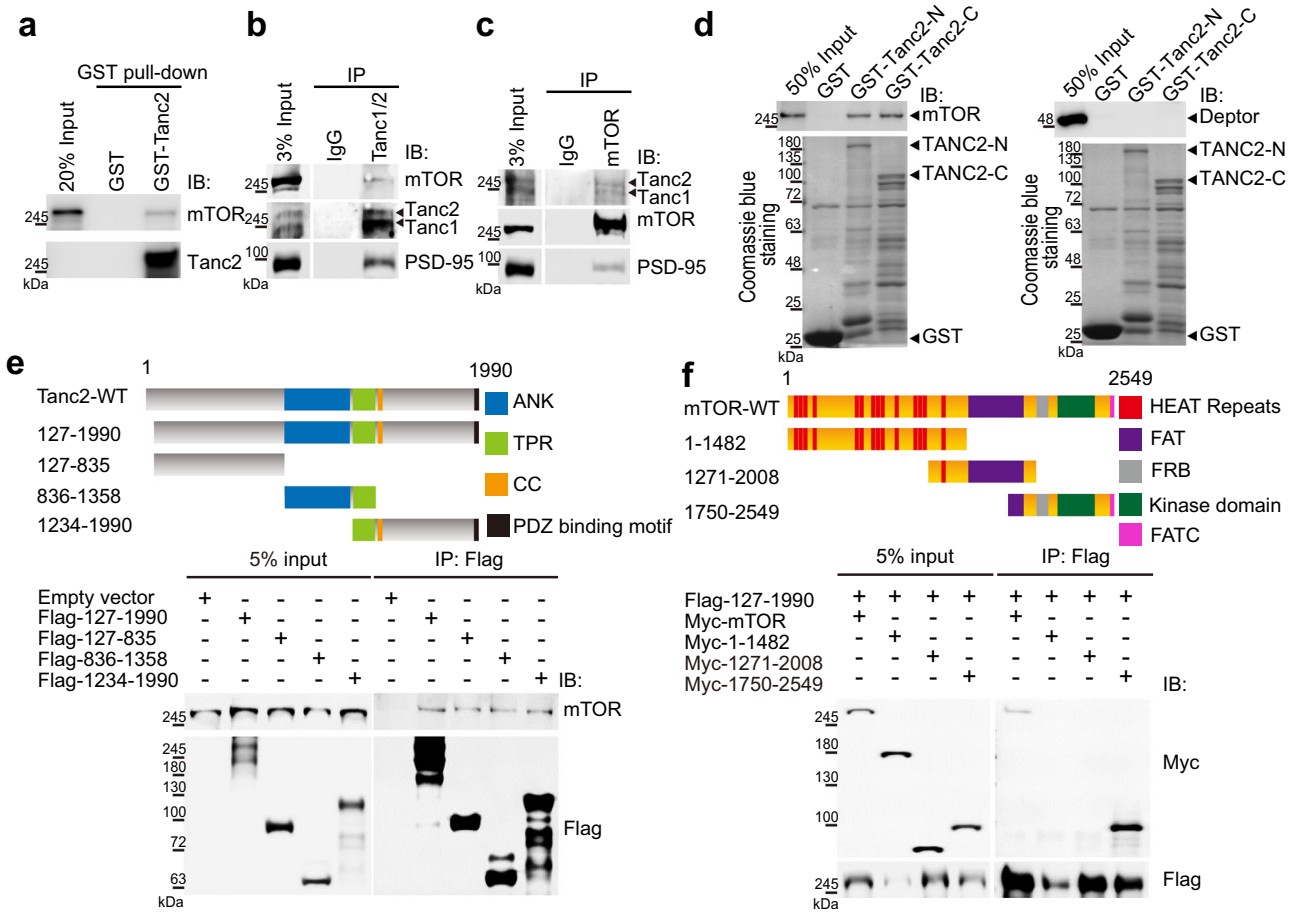

**Fig. 4 Tanc2 interacts with mTOR. a** Purified Tanc2 protein directly interacts with purified mTOR protein. GST-Tanc2 (full-length) protein was used to pull down purified mTOR protein. Input, 20%. Three independent experiments yielded similar results. **b, c** Tanc1 and Tanc2 form a complex with mTOR in the mouse brain. Whole-brain lysates (P14; mouse) were immunoprecipitated (IP) with pan-Tanc or mTOR antibodies, followed by immunoblotting. Note that mTOR pull-down also coprecipitated PSD-95 through Tanc2. Three independent experiments yielded similar results. **d** Both N- and C-terminal regions of purified Tanc2 protein directly interact with purified mTOR but not purified Deptor. GST-tagged purified N- and C-terminal regions of human Tanc2 (aa 1-1358 and aa 1359-1990) were coupled to glutathione beads and incubated with purified mTOR and Deptor proteins, followed by GST pull-down and immunoblot analysis. Three independent experiments yielded similar results. **e** Tanc2 forms a complex with mTOR in HEK293T cells through multiple regions of Tanc2. Lysates of HEK293T cells expressing deletion variants of Flag-Tanc2 (near-full-length, aa 127-1990; N-terminal region, aa 127-835; middle region, aa 836-1358; C-terminal region, aa 1234-1990) and mTOR (endogenous) were immunoprecipitated with Flag antibodies and immunoblotted with anti-Flag (for Tanc2) and mTOR antibodies. Note that all four deletion variants of Tanc2 interacted with mTOR, suggesting that multiple regions of Tanc2 are involved in mTOR binding. We used the near-full-length Tanc2 because the full-length construct was unavailable at the time of the experiment; experiments repeated using the full-length Tanc2 construct yielded the same results. ANK ankyrin repeats, TPR tetratricopeptide repeats, CC coiled-coil domain, PB PDZ-binding motif. Three independent experiments yielded similar results. **f** The C-terminal region of mTOR containing FRB, kinase, and FATC domains is sufficient for complex formation with Tanc2. HEK293T cells expressing Myc-mTOR deletion variants and Flag-Tanc2 (aa 127-1990) were immunoprecipitated with Flag antibodies and immunoblotted with Myc and Flag antibodies. FRB, FKBP12-rapamycin binding; FATC, FRAP-ATM-TRRAP-C-terminal domain. Three independent experiments yielded similar results.

and Deptor double-knockdown did not produce additive effects at early or late stages, except with respect to early-stage (DIV 7–14) mTOR phosphorylation (Fig. 7a–d). These results suggest that Tanc2 and Deptor/Tanc1 distinctly inhibit mTORC1/2 signaling at early and late stages of mouse brain development, respectively, in line with the embryonic lethality of *Tanc2*, but not *Deptor* or *Tanc1*, KO mice[14,22].

To determine whether neurons or glial cells are more important for Tanc2-dependent mTOR inhibition, we selectively knocked down Tanc2 in neuron- or glia-enriched early-stage cultured hippocampal neurons (DIV 7–14). Neuronal, but not glial, Tanc2 knockdown induced mTOR hyperactivity, and, in line with this, Tanc2 expression was much weaker in glial cells (Supplementary Fig. 8), suggesting that Tanc2 is more important for mTOR inhibition in neurons than in glial cells at early stages.

Within the neuronal populations, Tanc2 mRNAs were detected in both glutamatergic and GABAergic neurons in mouse brains at P7 and P14, as shown by fluorescent in situ hybridization (FISH) (Supplementary Fig. 9).

**TANC2 in human neurons inhibits mTORC1 and mTORC2.** Finally, we tested whether Tanc2 inhibits mTOR activity in human neurons. To this end, we knocked down *TANC2* in human neural progenitor cells (NPCs) developing into neurons for 2 weeks using two independent *TANC2* knockdown constructs. Both *TANC2* knockdown constructs similarly increased phosphorylation of S6 (S235/236), 4E-BP (T37/46), and GSK3β (S9), although they exerted moderate effects on Akt (S473) phosphorylation (Fig. 8a–c and Supplementary Fig. 10). mTOR phosphorylation was unaltered, similar to the results from mouse

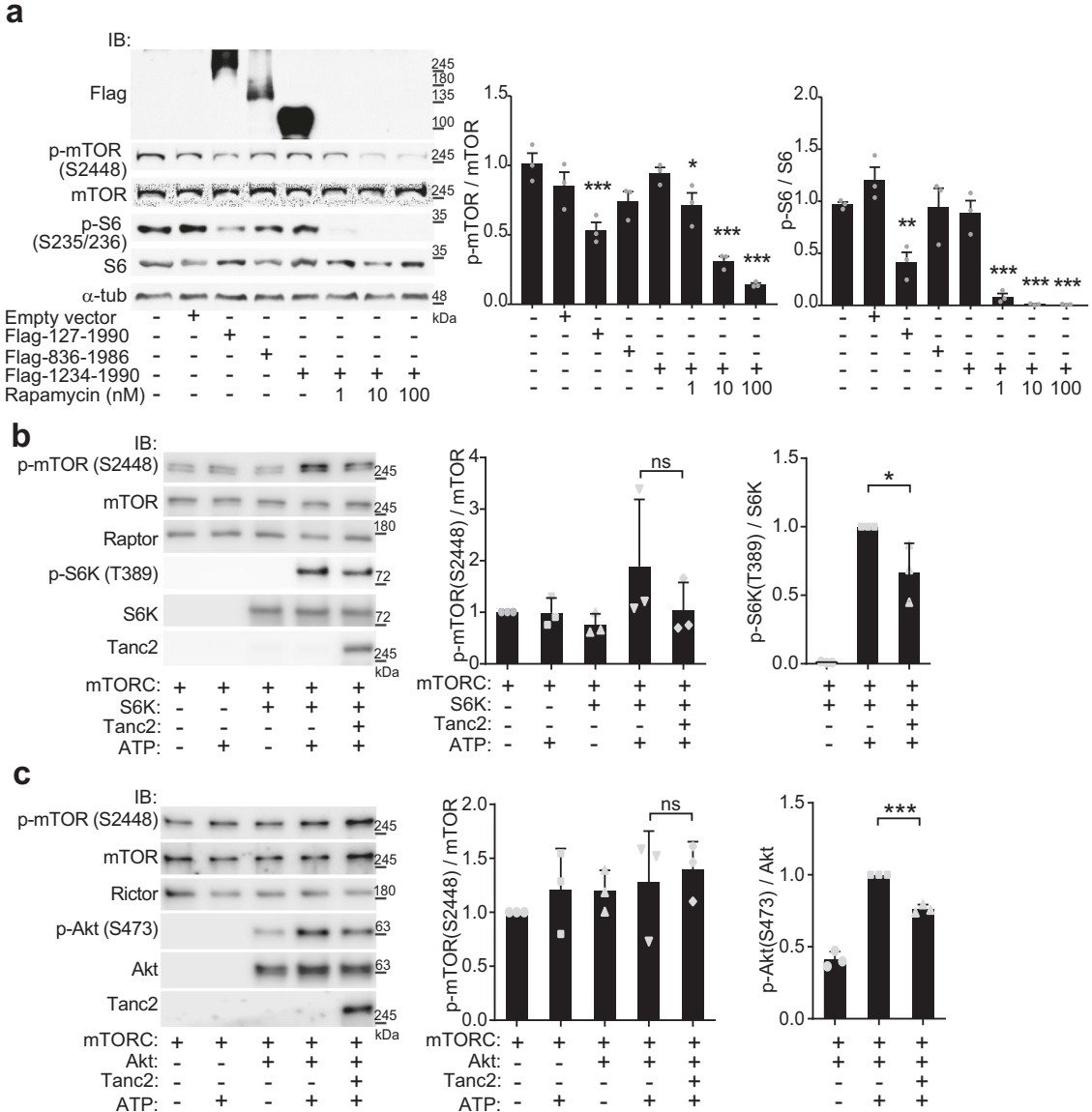

**Fig. 5 Tanc2 inhibits mTOR activity. a** Tanc2 suppresses endogenous mTOR activity in HEK293T cells. HEK293T cells were transfected with near-full-length Flag-Tanc2 (aa 127–1990; human) or Flag-Tanc2 deletion variants (aa 836–1986 and 1234–1990; human), followed by immunoblot analysis for phosphorylated (Ser-2448) and total mTOR. Cells in which mTOR was suppressed by rapamycin served as controls. Note that the near-full-length variant significantly inhibited mTOR activity, whereas the shorter deletion variants (aa 836–1986, aa 1234–1990) did not. Data: mean ± SD. ($n = 3$ independent experiments, *$P < 0.05$, **$P < 0.01$, ***$P < 0.001$ [compared with the first bar in each graph], one-way ANOVA with Bonferroni test). **b** Purified Tanc2 inhibits the kinase activity of purified mTORC1 (containing GFP-mTOR and mTOR-associated proteins such as Raptor), as shown by decreased phosphorylation of purified S6K (an mTOR substrate). In control experiments, EGFP-Tanc2 was replaced with purified EGFP protein (lanes 1–4; not probed). Data: mean ± SD. ($n = 3$ independent experiments, *$P < 0.05$ [compared to the absence of Tanc2/the second or fourth bar in the middle and right panels, respectively], ns not significant, one-way ANOVA with Bonferroni test). **c** Purified Tanc2 inhibits the kinase activity of purified mTORC2 (containing GFP-mTOR- and mTOR-associated proteins such as Rictor), as shown by decreased phosphorylation of purified Akt (an mTOR substrate). The baseline phosphorylation in Akt could be attributable to that a small portion of purified Akt proteins is phosphorylated or that the antibody recognizes non-phosphorylated proteins in addition to phosphorylated proteins. Data: mean ± SD. ($n = 3$ independent experiments, ***$P < 0.001$ [compared to the absence of Tanc2/the fourth bar], ns not significant, one-way ANOVA with Bonferroni test). See Source Data 1 for raw data values and Source Data 2 for statistical details.

neurons (Fig. 7). These results collectively suggest that Tanc2 inhibits mTORC1/2 in both human and mouse neurons.

## Discussion

The present study suggests that Tanc2 is a novel and regulated mTOR inhibitor that has strong neurodevelopmental impacts and therapeutic potential. The first important conclusion from our results is that Tanc2 binds to mTOR. In support of this,

Tanc2 forms a complex with mTOR in heterologous cells and in the mouse brain. Moreover, purified Tanc2 proteins form a complex with purified mTOR proteins. Tanc2 uses its multiple domains to associate with mTOR, whereas mTOR binds to Tanc2 through its C-terminal region, containing the FRB, kinase, and FATC domains. The latter is further supported by that rapamycin, known to bind to the FRB domain of mTOR, blocks the colocalization and biochemical association between Tanc2 and mTOR.

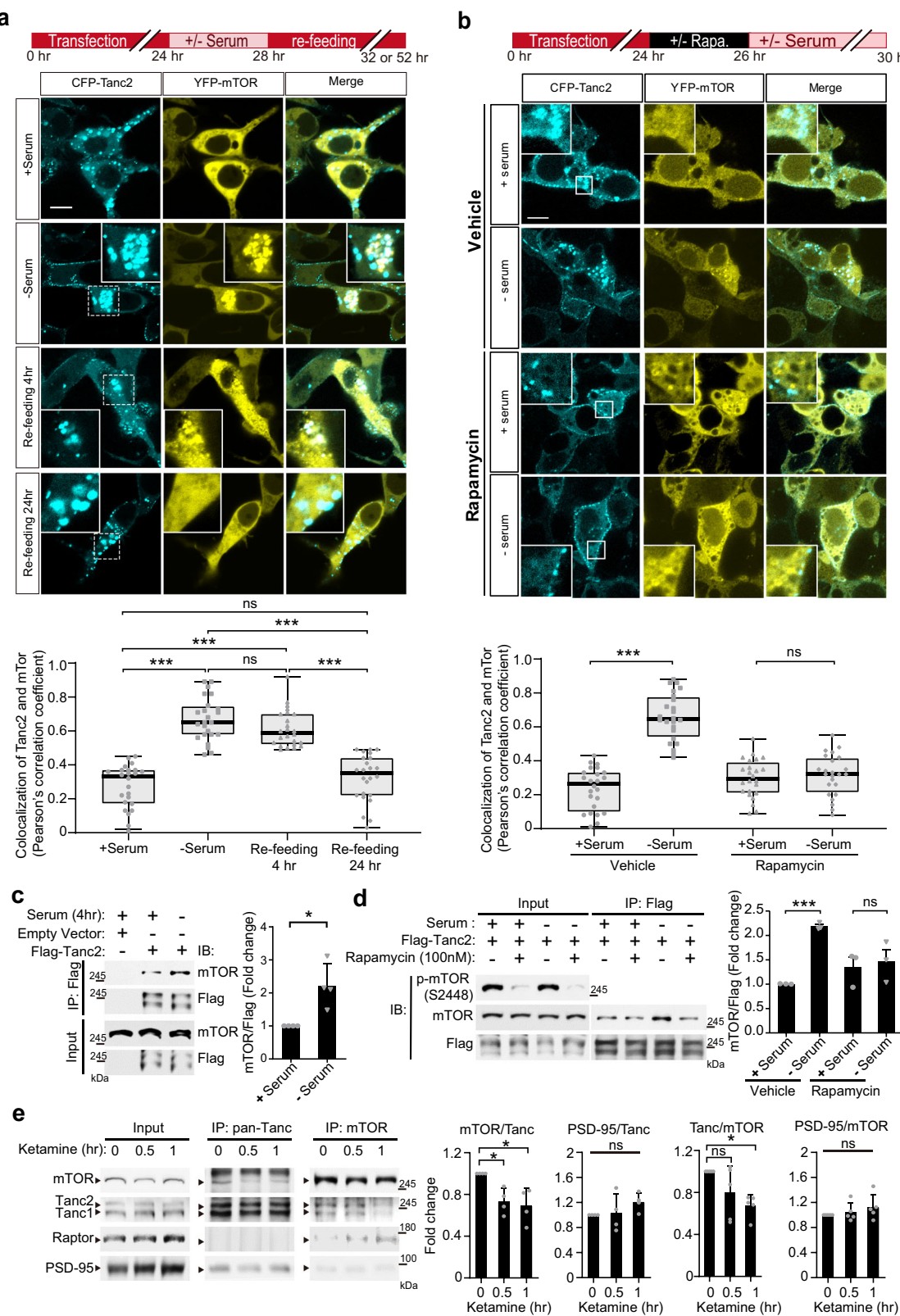

Tanc2 binds to mTOR in a regulated manner. The presence of serum, well known to activate mTOR, weakens the colocalization and biochemical association between Tanc2 and mTOR. In addition, ketamine, a fast-acting antidepressant known to promote excitatory synapse functions and mTOR activity[41], inhibits the Tanc2–mTOR interaction in the mouse brain. These results suggest that Tanc2 inhibits mTOR in a regulated manner to coordinate mTOR activity under various nutritional states and during brain development and neuronal or synaptic activities. Specific mechanisms that underlie the regulated Tanc2–mTOR interactions remain to be determined, although they could be posttranslational modifications of mTOR or Tanc2 at binding interfaces or regulatory domains.

**Fig. 6 Serum and ketamine regulate the interaction of Tanc2 with mTOR. a** Serum starvation induces colocalization of Tanc2 and mTOR in HEK293T cells, an effect that is reversed by serum refeeding. HEK293T cells transfected with CFP-Tanc2 + YFP-mTOR were subjected to serum starvation (-serum) to inactivate mTOR, or to no serum starvation (control; +serum), for 4 h followed by serum refeeding for 24 h while checking changes at 4- and 24-h time points. The colocalization was quantified using Pearson's correlation analysis of colocalized pixels (see "Methods" for details). Data: minimal, maximal, median, 25 and 75% values. ($n = 24$ cells from three independent experiments. ***$P < 0.001$, ns not significant, one-way ANOVA with Bonferroni test). Scale bar, 10 μm. **b** Rapamycin blocks serum starvation-induced Tanc2–mTOR colocalization in HEK293T cells. HEK293T cells expressing CFP-Tanc2 and YFP-mTOR were treated with rapamycin or vehicle for 2 h before starting serum starvation. Colocalization was quantified by Pearson's correlation analysis of colocalized pixels (see "Methods" for details). Data: minimal, maximal, median, 25 and 75% values. ($n = 24$ cells from three independent experiments. ***$P < 0.001$, ns not significant, Student's $t$ test). Scale bar, 10 μm. **c** Increased biochemical association between Tanc2 and mTOR induced by serum starvation in HEK293T cells, as determined by coimmunoprecipitation (coIP). HEK293T cells expressing Flag-Tanc2 and mTOR (endogenous) in the presence and absence of serum starvation (4 h) were immunoprecipitated with Flag antibody (for Tanc2) and immunoblotted as indicated. The lower Flag-Tanc2 band represents a degradation product. mTOR signals were normalized to Tanc2 signals for quantification. Data: mean ± SD. ($n = 4$ independent experiments, *$P < 0.05$, Student's $t$ test). **d** Rapamycin blocks the serum starvation-induced biochemical association of Tanc2 with mTOR in HEK293T cells, as determined by coimmunoprecipitation. HEK293T cells expressing Flag-Tanc2 and mTOR (endogenous) were treated with rapamycin or vehicle for 2 h before starting serum starvation, followed by immunoprecipitation with Flag antibodies (for Tanc2) and immunoblotting, as indicated. mTOR signals were normalized to Tanc2 (Flag) signals for quantification. Data: mean ± SD. ($n = 3$ independent experiments, ***$P < 0.01$, ns not significant, Student's test). **e** Reduced biochemical association between Tanc1/2 and mTOR in the mouse brain (P13–14) upon ketamine treatment (10 mg/kg; i.p.), as shown by coIP experiments on whole-brain crude synaptosomes from ketamine-treated and -untreated mice using pan-Tanc or mTOR antibodies, followed by immunoblotting. Raptor was immunoblotted to show mTOR activation, and PSD-95 was immunoblotted for the coprecipitation with Tanc1/2 (positive control). Data: mean ± SD. ($n = 4$ independent experiments, *$P < 0.05$ [compared with 0 h], ns not significant, one-way ANOVA with Bonferroni test). See Source Data 1 for raw data values and Supplementary Table 1 for statistical details.

Perhaps the most important conclusion of the present study is that Tanc2 inhibits mTOR. This is supported by multiple lines of in vitro and vivo evidence. Most directly, purified Tanc2 inhibits the kinase activity of mTOR, as shown by the suppression of mTORC1/2-dependent phosphorylation of mTOR substrates (S6K and Akt). In addition, Tanc2 overexpressed in HEK293T cells inhibits mTOR. Moreover, acute knockdown of Tanc2 increases mTOR activity in cultured mouse neurons at around, but not after, the developmental stages of strong Tanc2 expression (P7–14 but not P21–28). $Tanc2^{+/-}$ mice show increased mTOR activity in both mTORC1 and mTORC2 complexes at P14 but not at P28 or P52. In addition, Cre-dependent acute knockout of Tanc2 in an independent Tanc2-mutant mouse line during P5–14 (but not P19–28) increases mTOR activity in mTORC1/2. In human neurons, Tanc2 knockdown increases mTOR activity in mTORC1/2 in NPCs and neurons. These results collectively suggest that Tanc2 binds to and inhibits mTOR in mouse and human neurons at early stages.

In addition to Tanc2, Tanc1 interacts with and inhibits mTOR in a rapamycin-dependent manner. Tanc1 expression sharply increases during postnatal stages of mouse brain development, whereas Tanc2 expression is stronger at earlier stages. Deptor, a known mTOR inhibitor, also shows strong late-stage expression, similar to Tanc1. It is therefore possible that Tanc2, Tanc1, and Deptor distinctly inhibit mTOR across different developmental stages. Indeed, our results indicate that Tanc2 and Tanc1/Deptor inhibit mTOR more strongly at around postnatal weeks 2 and 4, respectively. These results are in line with the differential impacts of homozygous Tanc2 and Tanc1/Deptor deletions in mice, where the deletion of Tanc2, but not Tanc1 or Deptor, leads to embryonic lethality[14,22].

Tanc2 and Tanc1 interact with the PSD-95 family of scaffolding proteins, known to mediate the molecular organization of multi-protein complexes at cell-to-cell junctions such as neuronal synapses in order to couple receptor activations with signaling pathways[24,25]. Therefore, Tanc2 and Tanc1 may recruit mTORC1/2 complexes to PSD-95-based multiple protein complexes at excitatory postsynaptic sites. In line with this idea, Tanc2 has been suggested to recruit cargo dense core vesicles driven by the KIF1A motor protein to excitatory synapses[27]. Synaptically localized mTORC1/2 may be inhibited by local Tanc2 until mTOR activity is increased by the activation of synaptic receptors such as TrkB and mGluRs[42]. The four known members of the PSD-95 family (PSD-95, PSD-93, SAP102, and SAP97) display differential spatiotemporal expression patterns; i.e., PSD-95 and PSD-93 are more abundant at later developmental stages whereas SAP102 expression is stronger at earlier stages. It is therefore possible that Tanc2 and Tanc1 may coordinate mTORC1/2 signaling at both synaptic and non-synaptic sites of PSD-95-enriched multi-protein complexes in developing neural and nonneural tissues.

The synaptic and behavioral phenotypes of $Tanc2^{+/-}$ mice implicate Tanc2 in the regulation of synaptic plasticity and behaviors, including LTP, learning and memory, hyperactivity, and anxiety-like behavior, all of which are reversed by rapamycin-dependent mTOR inhibition. In humans, $TANC2$ mutations have been extensively associated with various neurodevelopmental and neuropsychiatric disorders, including intellectual disability, schizophrenia, and ASD[23,28–36]. These results, together with the embryonic lethality in $Tanc2^{-/-}$ mice and strongly increased mTOR activity in $Tanc2^{+/-}$ mice, suggest that Tanc2 regulates normal bran development and function by coordinating mTOR inhibition and that rapamycin-dependent mTOR inhibition could possibly be used to treat human patients with Tanc2 mutations and resulting mTOR hyperactivity. In addition, modulation of Tanc2 activity, i.e., antisense Tanc2 knockdown, might have therapeutic potential for mTOR-related brain disorders[5–7,43–45].

In conclusion, our study reports that Tanc2 is a regulated mTOR inhibitor with strong neurodevelopmental impacts and that mTOR inhibition could be an effective strategy for treating human individuals with $TANC2$ mutations suffering from neuropsychiatric disorders, including intellectual disability, ASD, developmental delays, and schizophrenia. Moreover, Tanc2 modulations promoting or suppressing mTOR signaling have therapeutic potential for the treatment of various mTOR-related peripheral and brain disorders.

## Methods
**Materials**. Reagents were obtained from the following sources: Pan-Tanc antibody has been previously described[22] and Tanc2-specific antibody made in-house; rabbit polyclonal antibodies to Akt (#9272), phosphor-Akt (#9271, #4060), Erk1/2 (#9102), phospho-Erk1/2 (#9101), GSK3β (#9315), phospho-GSK3β (#9336), PTEN (#9559), phospho-PTEN (#9549), PI3K (#4257), phospho-PI3K (#4228), TSC1 (#6935), TSC2 (#4308), phospho-TSC2 (#3617), mTOR (#2983), mTOR (#2972, used for immunoprecipitation), phospho-mTOR (#2971), S6 (#2217),

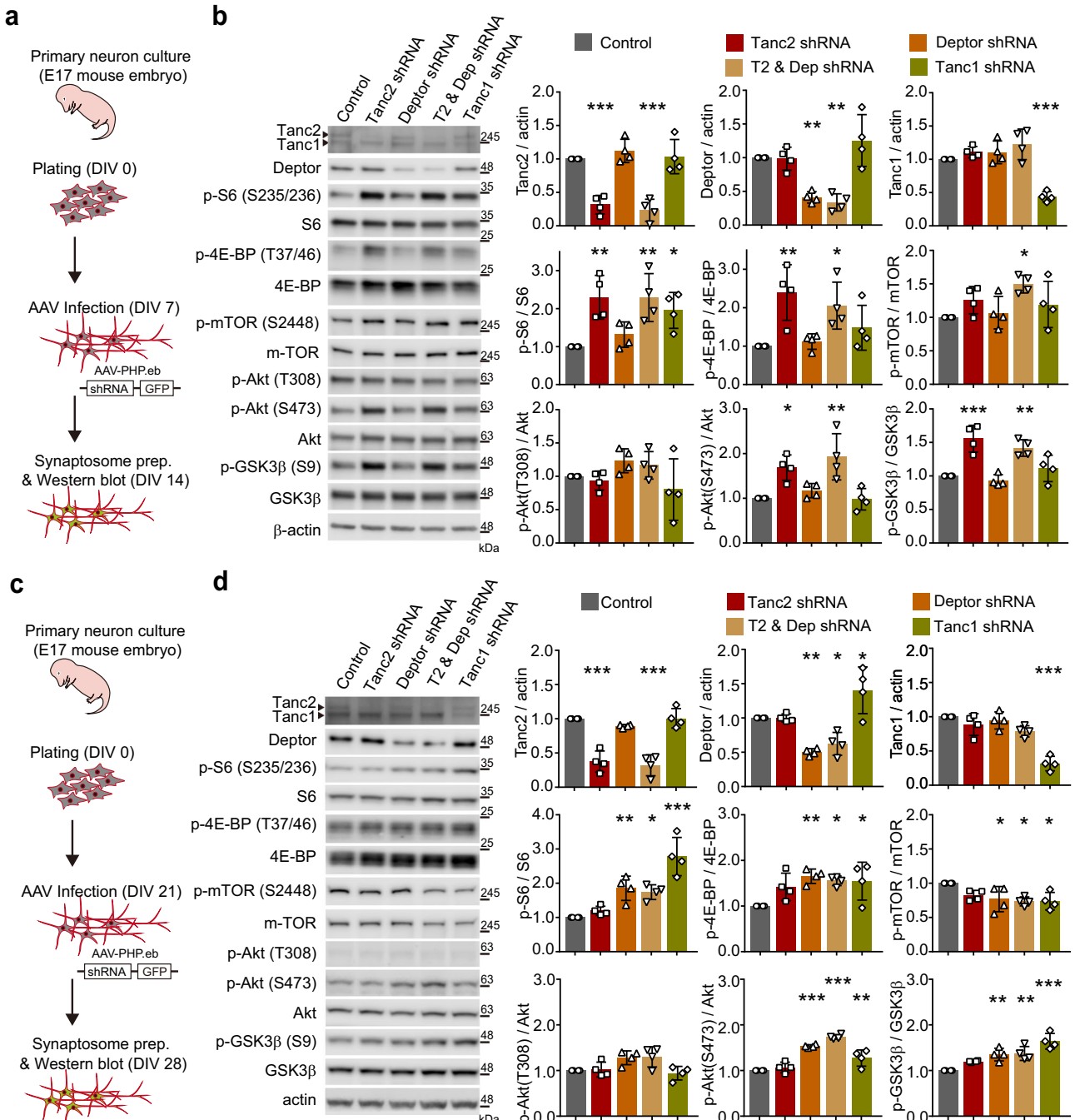

**Fig. 7 Acute knockdown of Tanc2, Deptor, and Tanc1 distinctly activate mTORC1/2 signaling in early- and late-stage neurons. a** Schematic depiction of AAV1-shRNA-mediated acute knockdown of Tanc2, Deptor, or Tanc1 in cultured mouse hippocampal neurons (DIV 7–14), and immunoblot analysis of crude synaptosomes for mTORC1/2 activity. **b** Early-stage (DIV 7–14), acute knockdown of Tanc2, but not Deptor or Tanc1, leads to hyper-phosphorylation of S6 (S235/236), 4E-BP (T37/46), Akt (S473), and GSK3β (S9), indicative of mTORC1 and mTORC2 hyperactivity. Note that knockdown of both Tanc2 (T2) and Deptor did not exert synergistic effects except for p-mTOR, and that Akt phosphorylation at T308 (not S473) was unchanged, indicative of unaltered activities of proteins upstream of mTOR. Data: mean ± SD. ($n = 4$ independent experiments; *$P < 0.05$, **$P < 0.01$, ***$P < 0.001$ [compared with control/the first bar in each graph], one-way ANOVA with Bonferroni test). **c** Schematic depiction of AAV1-shRNA-mediated knockdown of Tanc2, Deptor, or Tanc1 in cultured mouse hippocampal neurons (DIV 21–28) and immunoblot analysis of crude synaptosomes for mTORC1/2 activity. **d** Late-stage (DIV 21–28) acute knockdown of Deptor, or Tanc1, but not Tanc2, leads to hyper-phosphorylation of S6 (S235/236), 4E-BP (T37/46), Akt (S473), and GSK3β (S9), indicative of mTORC1 and mTORC2 hyperactivity. Data: mean ± SD. ($n = 4$ independent experiments; *$P < 0.05$, **$P < 0.01$, ***$P < 0.001$ [compared to control/the first bar in each graph], one-way ANOVA with Bonferroni test). See Source Data 1 for raw data values and Supplementary Table 1 for statistical details.

phospho-S6 (#4858), 4E-BP (#9644), phospho-4E-BP (#2855), S6K (#2708), phospho-S6K (#9205), and PRAS40 (#2691) from Cell Signaling Technology; Raptor (#09–217), phospho-Raptor (#09–107), phosphor-PKCa (#07–790), NeuN (Abn90), and Deptor (#abs222) antibodies (Millipore); PKCa (#610108) antibodies

(BD); Flag M2 (F1804), GFAP (G3893), and α-tubulin (T9026) antibodies (Sigma Aldrich); HA antibodies (M180-3) (MBL); GFP antibodies (sc-9996; Santa Cruz); HRP-labeled anti-mouse, anti-guineapig, and anti-rabbit secondary antibodies (Thermo Scientific); IRDye 800CW-conjugated and 680RD-conjugated goat anti-

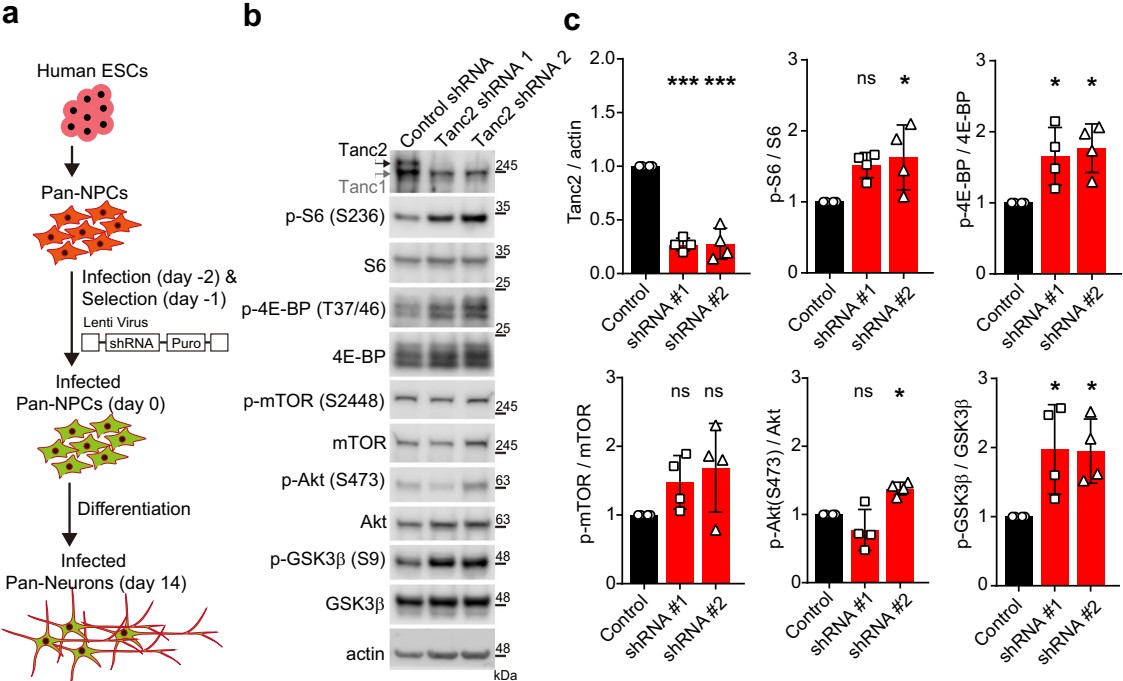

**Fig. 8 Acute TANC2 knockdown in human neurons induces hyperactivity of mTORC1 and mTORC2. a** Schematic of the experimental design to test if TANC2 knockdown increases mTORC1/2 activity in human neural progenitor cell (NPC)-derived neurons (day 14), which are less likely to be fully matured neurons at this stage. Pan-NPCs infected with lentivirus particles for TANC2 knockdown were selected and subjected to neuronal maturation for 2 weeks before the analysis of mTORC1/2 signaling by immunoblot analysis. **b, c** Knockdown of TANC2 in human neurons leads to hyper-phosphorylation of S6 (S235/236), 4E-BP (T37/46), and GSK3β (S9), indicative of increased mTORC1 and mTORC2 hyperactivity. Note that the effects of the two knockdown constructs are not identical, in particular, for Akt (S473) phosphorylation, which was unaltered and increased by the #1 and #2 knockdown constructs, respectively. It could be because of distinct properties of the two constructs, such as differences in the strengths of target sequence binding, time courses of target gene knockdown, or compensatory cellular responses to adjust Akt activity. Human NPCs infected with two independent TANC2 knockdown lentivirus particles (#1 and #2) were differentiated into neurons and analyzed by immunoblot analyses. Data: mean ± SD. ($n = 4$ independent experiments; $*P < 0.05$, $***P < 0.001$, ns not significant [compared to controls], one-way ANOVA with Bonferroni test). See Source Data 1 for raw data values and Supplementary Table 1 for statistical details.

mouse and goat anti-rabbit antibodies (LI-COR Biosciences); Dulbecco's Modified Eagle's Medium (DMEM) (Invitrogen); CHAPS detergent (Thermo Scientific); rapamycin (LC Labs); ketamine (Yuhan Co).

**DNA constructs.** The p3XFlag-Tanc1 and Tanc2 constructs were described previously[22]. Human Tanc2 deletion mutants (aa 127-835, 836-1358, and 1234-1990) were subcloned into p3XFlag-CMV-7.1 (Sigma). The full-length human Tanc2 gene was fully synthesized by overlapping PCR and subcloned into p3xFlag-CMV vector. The following constructs were purchased from Addgene: pAAV-CAG-GFP, pRK5-myc-mTOR, pRK5-myc-mTOR (aa 1-1482), pRK5-myc-mTOR (aa 1271-2008), pRK5-myc-mTOR (aa 1750-2549), pRK5-HA-Raptor, pRK5-HA-YFP-Deptor, pRK5-Flag-Deptor (DEP domains), pRK5-Flag-Deptor (PDZ domain). pEYFP-mTOR and pAAV-CAG-GFP-mTOR were generated using pRK5-myc-mTOR as template and subcloned into pEYFP-C1 (Clontech) and pAAV-CAG-GFP vector. pECFP-Tanc2 (aa 1-1990; human), pAAV-CAG-EGFP-Tanc2 (aa 1-1990), pGST-Tanc2 (aa 1-1990), pGST-Tanc2-N (aa 1-1358), and pGST-Tanc2-C (aa 1359-1990) were generated using p3xFlag-Tanc2 (aa 1-1990) as template and subcloning into pECFP-C1 (Clontech), pAAV-CAG-EGFP (Addgene #28014), and pGEX-6P-1 (GE Healthcare) vectors. pGST-Deptor was generated using pRK5-flag-Deptor (Addgene #21334) as template and subcloning into pGEX-6P-1 vector. Mouse Tanc1 shRNA (target sequence:5'-cgaagaaaggagcagagata), Tanc2 shRNA (target sequence:5'-gctcaga gatccagaataa), and Deptor shRNA (target sequence: 5'-cgcaaggaagacattcacgat) were subcloned into pAAV-U6-GFP vector. Human Tanc2 shRNA#1 (target sequence: 5'-gctcagagatccagaataa) and shRNA#2 (target sequence: 5'-ggacagagaggttttaaataa) were subcloned into pLKO.1 (Addgene #10878) vector. All constructs were verified by DNA sequencing (Cosmo Genetech).

**Fluorescent in situ hybridization (FISH).** In brief, frozen sections (14-μm thick) were cut coronally through the hippocampal formation. Sections were thaw-mounted onto Superfrost Plus Microscope Slides (Fisher Scientific #12-550-15). The sections were fixed in 4% paraformaldehyde for 10 min, dehydrated in increasing concentrations of ethanol for 5 min, and finally air-dried. Tissues were then pretreated for protease digestion for 10 min at room temperature. For RNA

detection, incubations with the different amplifier solutions were performed in a HybEZ hybridization oven (Advanced Cell Diagnostics, Hayward) at 40 °C. The probes used in these studies were three synthetic oligonucleotides complementary to the sequence 289–1198 of Mm-Tanc2, the sequence 62–3113 of Mm-Gad1-C3, the sequence 552–1506 of Mm-Gad2-C3, the sequence 464–1415 of Mm-Slc17a7-C2, and the sequence 1986–2998 of Mm-Slc17a6-C2 (Advanced Cell Diagnostics, Hayward), respectively. The labeled probes were conjugated to Alexa Fluor 488, Atto 550, and Atto 647. The sections were hybridized 2 h at 40 °C with labeled probe mixture per slide. Then the nonspecifically hybridized probe was removed by washing the sections, there times for 2 min each in 1X wash buffer at room temperature. Then Amplifier 1-FL (30 min), Amplifier 2-FL (15 min), Amplifier 3-FL (30 min), and Amplifier 4 Alt B-FL (30 min) were sequentially applied for 15 min at 40 °C. Each amplifier solution was removed by washing with 1X wash buffer for 2 min at room temperature. Fluorescent images were acquired using TCS SP8 Dichroic/CS (Leica), and the ImageJ program (NIH) was used to analyze the images.

**AAV production and injection.** AAV1-hSyn-Cre-eGFP (pENN.AAV.hSyn.HI. eGFP-Cre.WPRE.SV40) and AAV1-hSyn-ΔCre-eGFP (pENN.AAV.hSyn.eGFP. WPRE.bGH) were a gift from James M. Wilson (Addgene #105539-AAV1, #105540-AAV1). For the preparation of AAV particles, we used HEK293T cells. For a single virus preparation, three of 150-pi dishes were used. When cells grew up 90% confluency, 10 μg of target plasmid, 20 μg of PHP.eB plasmid, and 10 μg of pAAV-helper plasmid were co-transfected in a 150-pi dishes using poly-etherimide (PEI) (Polysciences, #23966-1) transfection method with the N:P ratio of 25. Twenty-four hours after transfection, the medium was changed with fresh DMEM + 5% fetal bovine serum (FBS). Seventy-two hours after transfection, the medium was collected at 4 °C for the next step, and replaced with fresh DMEM + 5% FBS. After 120 h post-transfection, the medium and cells were harvested together and separated by centrifugation. The supernatant was mixed with 1/5 volume of 40% w/v PEG 8000 (Sigma, #89510)/2.5 NaCl solution. After 2 h of incubation on ice, the mixtures were centrifuged at $4000 \times g$ for 30 min. Pellets were suspended in 6-ml SAN digestion buffer with 100-U/ml SAN (HL-SAN, Arcticzymes, #70910-202; Buffer: 23-mM Tris-HCl, pH = 8.5/5-mM MgCl₂/0.5-M

NaCl) and incubated at 37 °C for 1 h. AAV virus particles were separated by iodixanol gradient (Optiprep; Sigma D1556) by ultra-centrifugation at 350,000 × $g$ for 2 h. Collected virus particles from the 42/60% iodixanol interface were diluted with Dulbecco's phosphate-buffered saline (DPBS) and dialyzed using Amicon Ultra-15 with 100-kDa cutoff. Injection of AAV1-hSyn1-Cre/ΔCre-GFP into the mouse hippocampus was performed using stereotaxic apparatus (Kopf Instruments) and mice anesthetized with isofluorane (Piramal Healthcare). Virus solutions were injected into the hippocampus dCA3 region in mice at postnatal day or P5 (AP: Lambda + 1.7, ML: ± 1.7, DV: −1.9) at the speed of 100 nL/min. To determine injection sites more precisely with naïve eyes, AAV1-hSyn-ΔCre-eGFP virus with stronger eGFP fluorescence was mixed and coinjected with AAV1-hSyn-Cre-eGFP at 1:3–1:5 ratios. Virus-injected brains were sliced (100-μm thickness) using a vibratome (Leica VT1200) in ice-cold dissection buffer (212-mM sucrose, 25-mM NaHCO₃, 5-mM KCl, 1.25-mM NaH₂PO₄, 0.5-mM CaCl₂, 3.5-mM MgSO₄, 10-mM D-glucose, 1.25 L-ascorbic acid, 2-mM Na-pyruvate). At P14, five slices were collected for GFP control and Cre-GFP, which were further dissected for GFP-positive regions. The dissected tissues were fractionated for crude synaptosome preparation and then subjected to immunoblotting.

**GST protein purification.** Expression constructs for GST alone, GST-Tanc2, GST-Tanc2-N (aa 1-1358), GST-Tanc2-C (aa 1359-1990), and GST-Deptor were transformed into BL-21 (DE3) *E.coli.* Cells were cultured in 2 x YT medium with ampicillin until OD600 reached 0.6 at 37 °C, and then protein expression was induced with 0.5-mM IPTG for 6 h at 25 °C. Cells were lysed in lysis buffer (1% Triton X-100, 150-mM NaCl, 20-mM Tris pH 7.4, 1-mM MgCl₂, 1-mM EGTA, 1-mM PMSF) with sonication and centrifuged at 20,000 x $g$ for 15 min at 4 °C. The supernatant was incubated with glutathione sepharose 4B beads (GE Healthcare) for 1 h. Immobilized Deptor protein was eluted by GST-tag cleavage using Pre-Scision protease (GE healthcare, #27084351).

**Tanc2 and mTORC1 protein purification in HEK293T cells.** GFP, GFP-Tanc2, GFP-mTOR + HA-Raptor, or GFP-mTOR + Myc-Rictor were overexpressed in HEK293T cells by transfection using Lipofectamine 3000 for 48 h. GFP- or GFP-Tanc2-expressing cells were lysed with RIPA buffer, and the lysates were incubated with anti-GFP antibodies trapped in agarose beads (Chromotech, GFP-Trap_A) for 2 h at 4 °C. The immunoprecipitated complexes were washed three times with the RIPA buffer, and the immobilized proteins were eluted by 0.2-mM glycine-HCl, pH 2.5 for the in vitro kinase assay. For mTORC1 and mTORC2 purification, GFP-mTOR + HA-Raptor (or Myc-Rictor) expressed in HEK293T cells were stimulated with 10-μg/ml insulin (Sigma #I9278) for 30 min prior to lysis in mTOR lysis buffer (40-mM HEPES, 2-mM EDTA, 10-mM β-glycerophosphate, pH 7.4, and 0.3% CHAPS). The lysates were incubated with anti-GFP antibodies trapped in agarose beads for 2 h at 4 °C, and then sequentially rinsed with low salt mTOR wash buffer (40-mM HEPES, 150-mM NaCl, 2-mM EDTA, 10-mM β-glycerophosphate, pH 7.4, 0.3% CHAPS), high salt mTOR wash buffer (40-mM HEPES, 400-mM NaCl, 2-mM EDTA, 10-mM β-glycerophosphate, pH 7.4, 0.3% CHAPS), and mTOR wash buffer (25-mM HEPES, 20-mM KCl, pH 7.4).

**In vitro direct binding test.** GST, GST-Tanc2-N, GST-Tanc2-C, or GST-Tanc2 (full-length) proteins immobilized to agarose beads were incubated with the mTOR recombinant protein (0.5 μg; Origene #TP320457), or Deptor protein, in RIPA buffer for 2 h at 4 °C. Protein complexes in the agarose beads were washed with RIPA buffer three times and subjected to SDS-PAGE for Coomassie blue staining or immunoblotting.

**In vitro mTOR kinase assay.** For mTORC1 in vitro kinase assay, GTP-bound Rheb (Rheb-GTP) was prepared by mixing Rheb protein (2.5 μg; Origene #TP300307), GTPγS (0.1 mM; Sigma #G8634), and EDTA (10 mM), and incubating the mixture at 30 °C for 15 min, followed by the addition of 20-mM MgCl₂. Then, mTORC1 immobilized in beads, Rheb-GTP (100 ng), and GFP/GFP-Tanc2 (500 ng) were mixed in mTOR kinase assay buffer (25-mM HEPES, 20-mM KCl, 10-mM MgCl₂) and incubated for 20 min on ice prior to the start of the kinase assay. The enzyme reaction was initiated by adding the mTOR assay start buffer (10 μl, 25-mM HEPES, 10-mM MgCl₂, 140-mM KCl) with or without ATP (500 μM) and S6K (200 ng, mTORC1 kinase substrate) (Origene #TP317324), followed by incubation at 30 °C for 60 min. The enzyme reaction was stopped by adding the 4x SDS-PAGE sample buffer. For mTORC2 in vitro kinase assay, mTORC1 immobilized in beads and GFP alone/GFP-Tanc2 (500 ng) were mixed in the mTOR kinase assay buffer and incubated for 20 min on ice prior to the start of the kinase assay. The enzyme reaction was initiated by adding the mTOR assay start buffer (10 μl, 25-mM HEPES, 10-mM MgCl₂, 140-mM KCl) with or without ATP (500 μM) and inactive Akt (200 ng, mTORC2 kinase substrate) (EMD Millipore #14-279), followed by incubation at 30 °C for 60 min. The enzyme reaction was stopped by adding the 4x SDS-PAGE sample buffer.

**Cell culture and transfection.** HEK293T cells obtained from the American Type Culture Collection were cultured in DMEM medium supplemented with 10% (v/v) FBS (Gibco) and 1% (v/v) penicillin/streptomycin at 37 °C in a humidified 10% CO₂ atmosphere. For live imaging, 1.5 × 10⁴ cells were plated in each well of

μ-Plate 96 plates. Cells were transfected with Lipofectamine (Invitrogen) according to the manufacturer's instructions. After 24 h, adding the transfection mixture into a well, colocalization analysis proceeded. For immunoprecipitation, 1 × 10⁶ cells were plated in 60-mm dish and transfected as described above.

**Cell lysis and immunoprecipitation.** All cells were rinsed with ice-cold phosphate-buffered saline (PBS) before lysis. All cells exceptor for those used to isolate mTOR-containing complexes were lysed in the RIPA buffer (10-mM HEPES [pH 7.4], 150-mM NaCl, 1-mM EDTA, 0.1-mM MgCl₂, 1% nondiet P40 (NP40), 20-mM NaF, 1-mM NaVO₃, 1-μg/ml aprotinin, 1-μg/ml pepstatin, 10-μg/ml leupeptin, and 1-mM PMSF). The cell lysates were clarified by centrifugation at 20,000 × $g$ for 10 min, and the supernatant was collected. For Immunoprecipitations, cells were lysed with the RIPA buffer containing 0.3% CHAPS instead of 1% NP40. The lysates were incubated with primary antibodies at 4 °C overnight, incubated with 50% slurry of protein A or G agarose beads (Incosharm) for 1–3 h at 4 °C, and washed three times in lysis buffer. Immunoprecipitated proteins were denatured by adding 20 μl of sample buffer and boiling for 5 min, and subjected to SDS-PAGE.

**Immunoblot analysis.** Protein concentrations were determined using the Pierce BCA protein assay kit (Thermo Scientific) according to the manufacturer's protocol. Proteins resolved in SDS-PAGE were transferred to PVDF membrane (Millipore). After blocking with 5% nonfat milk or 5% BSA, the blots were incubated with primary antibodies overnight at 4 °C, incubated with HRP- or IRDye-conjugated secondary antibodies for 1 h at room temperature, and detected by using enhanced chemiluminescence (Thermo Scientific) or Odyssey CLx imaging system (LI-COR Biosciences). All of the uncropped full-length images can be found in Source Data 2.

**Live imaging and colocalization analysis.** Serum starvation and refeeding experiment were performed by plating HEK293T cells on 96-well plates and transfecting the cells as described above. After 24 h of transfection, cells were washed with PBS and maintained with serum-free culture medium at 37 °C in a humidified 10% CO₂ air for 4–5 h, then the medium was replaced with normal culture medium supplemented with serum, incubated for following 4 or 24 h. For rapamycin treatment, transfected cells were treated first with 100 nM of rapamycin or vehicle (4% ethanol, 4% PEG400, 4% Tween 80, and sterile water) for 2 h before the serum starvation procedure described above. Live-cell imaging for colocalization was performed using a Nikon A1R confocal microscope (Nikon Instruments) mounted in a Nikon Eclipse Ti body, and equipped with CFI Plan Apochromat VC objectives (60×/1.4-NA oil or 40×/0.95-NA air; Nikon) and a Chamlide TC system (maintained 37 °C and 10% or 10% CO₂; Live Cell Instrument, Inc). CFP and YFP images (512 × 512 pixels, 72.7 μm²) were taken using 457 and 514-nm laser lines, respectively. Colocalization images were quantified using ImageJ coloc2 plugin.

**Ketamine treatment and synaptosome preparation.** Mice received a single acute i.p. injection of ketamine (10 mg/kg, i.p.) and were sacrificed at the indicated times. Brain tissues were collected for crude synaptosomal preparation. To purify crude synaptosomes, brain tissues (without cerebellum) from P13–14 mice were rapidly removed and homogenized in a solution containing 0.32-M sucrose, 20-mM HEPES (pH 7.4), 1-mM EDTA, 20-mM NaF, 1-μg/ml aprotinin, 1-μg/ml pepstatin, 10-μg/ml leupeptin, 1-mM PMSF, and 1-mM Na₃VO₄. The homogenate was centrifuged at 900 × $g$ at 4 °C for 10 min. The pellet (P1, nuclear fraction) contains nuclei and large cell debris. The supernatant was centrifuged at 10,000 × $g$ at 4 °C for 10 min. After centrifugation, the supernatant (cytosolic fraction) was removed, and the pellet (P2: crude synaptosomal fraction) was resuspended and lysed in the RIPA buffer containing 0.3% CHAPS. Subsequently, P2 lysates were centrifuged for 15 min at 20,000 × $g$ at 4 °C to remove insoluble debris.

**Mice.** $Tanc2^{+/-}$ mice have been previously described (genetic background: C57BL/6J)[22]. All mice used in this study were generated by in vitro fertilization. To generate $Tanc2^{fl/fl}$ mice, $Tanc2^{tm2a}$(KOMP)Wtsi ES cells were obtained from KOMP. The mutant $Tanc2$ allele was inserted in the intron located between exons 4 and 5 of the $Tanc2$ gene. Embryonic stem cells were injected into blastocysts to generate chimeric mice. Chimeric mice were further bred to C57BL/6J WT mice to generate F1 heterozygous $Tanc2^{tm2a}$(KOMP)Wtsi mice ($Tanc2^{null/+}$). F1 mice were then bred with Protamine-Flp mice (C57BL/6J0 to remove the Frt-flanked gene trap cassette in the Tanc2 tm2a(KOMP)Wtsi allele, resulting in a conditional allele ($Tanc2^{fl/+}$). Homozygous $Tanc2^{fl/fl}$ mice were generated from $Tanc2^{fl/+}$ intercrosses, which allow further conditional knockout in the presence of Cre expression. $Tanc2^{fl/fl}$ mice were identified by PCR with primers (Supplementary Table 2) to amplify a 222-bp fragment from the WT allele and a 440-bp fragment from the flox allele. Mice maintenance and procedures were performed in accordance with the Requirements of Animal Research at KAIST. Experimental procedures were approved by the Committee on Animal Research at KAIST (KA2016-31). Mice were fed ad libitum, and 2–5 animals were housed in a cage with a 12-h dark/12-h light cycle.

**Behavioral tests.** All behavioral tests were performed using age-matched mice (2–4 months) during the dark phase of the cycle (light-off periods). Mice had at

least 24 h of rest time between tests. All experimental data were analyzed using Ethovision XT 10.1 software (Noldus) unless stated otherwise analysis. All tests were conducted in a blind manner.

*Open-field test.* Subjects were placed in a white acryl open-field box ($40 \times 40 \times 40$ cm), and the movements were recorded for 60 min for adult mice. Illuminations were set at 0 or 100 lux. The distance moved and time spent in the center arena (central area of $20 \times 20$) were measured.

*Laboras test.* Various behaviors, including locomotion, immobility, climbing, grooming, and rearing, were recorded automatically using Laboratory Animal Behavior Observation Registration and Analysis System (LABORAS$^{TM}$) by Metris[46,47]. Mice were housed in LABORAS recording cages. The cage is directly placed onto the sensing platform with the upper part of the cage, including the top, food hopper, and drinking bottle. The recording was conducted for 72 consecutive hours. Animals were tested at the age of 2–3 months.

*Morris water-maze test.* Morris water-maze testing was conducted in a round white pool 100 or 120 cm in diameter and 40-cm deep. The maze was placed alone in a room with four cues on its walls. The pool was filled with water until the top of the platform (10 cm in diameter) was submerged 1 cm below the water surface, and white paint was added. The pool temperature was maintained at $22 \pm 0.5$ °C by adding warm water. Each mouse was given 3 trials a day for 5 consecutive days. In each trial, mice were placed in three different quadrants in random order. Irrespective of the trial performance, mice were guided to the platform and allowed to remain there for at least 15 s. In the probe test performed 24 h after the acquisition task, mice were placed in the center of the arena and allowed to swim for 1 min in the absence of the platform. The number of times the mice crossed the learned escape platform location, as well as the time spent in the quadrant, was measured. On the day after the probe test, mice were subjected to a reversal task in which the platform was placed in the quadrant opposite side of the arena and received three trials a day during 3 consecutive days. The procedure remained the same as that of the acquisition task.

*Novel-object recognition test.* Each mouse was habituated in the open-field box without objects for 30 min a day before the training session. At the training session, the mouse was placed in the open-field arena containing two identical sample objects for 10 min. Twenty-four hours after the training, the mouse was returned to the open-field arena with two objects, one is identical to the sample, and the other is novel. Object recognition was scored by the amount of time with the nose of the mouse pointed and located within 2 cm from the object.

*Contextual fear conditioning test.* Mice at 4–5 months of age were placed in the conditioning chamber (Coulbourn instruments) and, after a 2-min adaptation period, received three foot shocks (2 s, 0.5 mA) at 2, 3, and 4 min. After the foot shock, mice remained in the chamber for an additional minute and then were returned to their home cage. After 24 h, mice were placed back into the same context for 5 min, and freezing was monitored. Data acquisition and analysis were measured using FreezeFrame (Coulbourn Instruments).

*Elevated plus-maze test.* The apparatus used for the elevated plus-maze test consisted of two open and closed (30-cm walled) arms ($5 \times 30$ cm each). The maze was elevated to a height 50 cm above the floor. Each mouse was placed in the central zone of the maze facing an enclosed arm and was recorded for 10 min. The frequency of entries and the amount of time spent in each arm were measured.

*Light–dark test.* The apparatus ($40 \times 20 \times 20$ cm) used for light–dark test was divided into two compartments; a smaller dark chamber that occupied roughly 1/3 of the total box, and a larger light chamber (200 lux), which was lit with a bright white light. There was a hole in between that allows mice to move freely between the compartments. The task ran for 10 min, and started after the mice were placed into the dark chamber. The anxiety parameters, time spent in the light compartment, and the number of entries into the light compartment were measured.

*Three-chamber social interaction test.* The three-chambered social approach test, originally developed for social interaction of rodents[48,49], was conducted as previously described[50]. In brief, plastic containment cups in the corner of both side chambers were placed. Mice were habituated in the center chamber for 10 min for adaptation, followed by in all three chambers for 10 min. Following habituation, Stranger 1 mouse (129/SvJae strain) was placed in a small container located in the corner of one side chamber, and the Object was placed in another container in the opposite side chamber. Subject interaction was recorded for 10 min. Following the first test, the Object was replaced with Stranger 2 (129/SvJae) and subject interaction was recorded for 10 min.

*Forced-swim test.* The apparatus, a glass beaker (15-cm diameter, 22-cm high), was filled with water (~24 °C) to the height of 15 cm. The time spent floating on the water (immobility time, s) during 6 min was manually scored.

*Tail-suspension test.* Mice were individually suspended by their tails to a horizontal metal bar using adhesive tape. The distance between the tip of the nose of the mouse and the floor was 20 cm. The time spent immobile was recorded for 6 min and manually scored.

**Field recordings.** Acute hippocampal slices (300-μm thick, sagittal slice for CA1) were used for extracellular field recordings at 27–30 °C (TC-324B, Warner Instruments). After a stable baseline was recorded, LTP was induced by HFS (100 Hz, 1 s for HFS-LTP). LTD was induced by DHPG (50 mM, 10 min for mGluR-LTD) or LFS (1 Hz, 900 stimuli for LFS-LTD) in the presence of picrotoxin (50 μM). Field EPSPs were evoked with a glass pipette containing external ACSF to activate SC-CA1 pyramidal synapses. Average responses (mean ± s.e.m.) were expressed as percentage of baseline response (at least 20 min of stable responses). For input/output relationship and paired-pulse ratio, 0.15-mV stimuli were given at 20-s intervals to acquire baselines. After stabilizing the baseline showing fiber volley amplitudes of 0.15–0.2 mV and rising slopes of 0.3–0.5 mV/ms, a protocol for each recording was given. Three sweeps were recorded for each stimulus intensity with increasing steps of 0.05 mV, starting from 0 mV until fiber volley reached 0.5 mV. For the paired-pulse ratio, two consecutive stimulus with 25, 50, 75, 100, 200, and 300-ms intervals were given. Three sweeps were recorded per interval with inter-experimental intervals of 30 s. For input–output ratios, rising field EPSP slopes were plotted against fiber volley amplitudes.

**Rapamycin rescue.** Rapamycin was freshly dissolved in a vehicle made of 4% ethanol, 4% PEG400, 4% Tween 80, and sterile water before use. For the $Tanc2^{+/-}$ in vivo experiments, vehicle or rapamycin was administered intraperitoneally on the same 3 days every week for 1 month (i.e., every Monday, Wednesday, and Friday) with rapamycin daily at a dose of 5 mg/kg starting at P10.

**Neuron culture and knockdown using AAV1 shRNAs.** Cultured mouse neurons were prepared from fetal C57/BL6J mice at embryonic day 17. Briefly, dissected hippocampal tissues were digested with papain and plated on poly-D-lysine-coated 6-well culture plates with a plating medium (Neurobasal-A medium supplemented with 2% B-27, 10% FBS, 1% GlutaMax, and 1-mM sodium pyruvate, all from Thermo Fisher Scientific) at the density of $6 \times 105$ cells per well. Four hours after plating, the plating medium was replaced with FBS-free culture medium (Neurobasal-A medium supplemented with 2% B-27, 1% GlutaMax, and 1-mM sodium pyruvate) with following 50% replacements every 7 days. On DIV 7 or 21, AAV shRNA particles were added to neurons and incubated for 7 days. Then, virus-infected cells were rinsed with ice-cold Tyrode's solution (136-mM NaCl, 2.5-mM KCl, 2-mM CaCl$_2$, 1.3-mM MgCl$_2$, 10-mM Na-HEPES, 10-mM D-Glucose, pH 7.3). For crude synaptosome preparation, AAV-infected neurons were collected in HEPES-buffered sucrose solution (0.32-M sucrose, 4-mM HEPES, pH 7.4, 1-mM EDTA, 20-mM NaF, 1-μg/ml aprotinin, 1-μg/ml pepstatin, 10-μg/ml leupeptin, 1-mM PMSF, and 1-mM Na$_3$VO$_4$) and homogenized using Glass-Teflon homogenizer and centrifuged for 10 min at $900 \times g$ at 4 °C. The supernatants were centrifuged for 10 min at $10,000 \times g$ at 4 °C. After centrifugation, the supernatant (cytosolic fraction) was removed, and the pellet (P2: crude synaptosomes) was resuspended and lysed in 1 x SDS SDS sample buffer for immunoblotting.

**Neuron and glia segregation culture.** Cytosine arabinoside (1 mM, Ara-C, Sigma #C1768) was added to FBS-free culture medium during DIV3–14 for glia-free neuron culture, and cytosine arabinoside was not added to 10% FBS-containing plating medium for glial cell culture. AAV-PHP.eB-shRNA particles were added to cultured neurons or glia at DIV 7, followed by cell harvest at DIV 14 for synaptosome preparation from neurons or total lysate preparation for glia.

**Lentivirus production.** Lentivirus was packaged in HEK293T cells grown in DMEM with 10% FBS. HEK293T cells were transfected with polyethyleneimine (PEI) (Polysciences #23966). Per 150-mm plate, the following solution was prepared, incubated for 5 min at room temperature and added drop-wise to plates: 12.2-μg lentiviral DNA, 8.1-μg MDL, 3.1-μg Rev-RSV, 4.1-μg CMV-VSVG, 1 ml of Opti-MEM (Thermo Fisher #31985062), and 110-μl PEI (1 μg/μl) and mixed lightly. The culture medium was changed after 4 h, and the virus was harvested at 72 h post transfection. To collect viruses, the cell medium was collected and centrifuged at $28,000 \times g$ for 2 h. The virus pellet was resuspended with DPBS. All the lentiviral vectors used contained the puromycin-resistance gene.

**Human neuron culture and lentivirus infection.** HUES6 obtained from HSCI iPS Core was maintained on MEF (ATCC, #SCRC-1040) in human embryonic stem cell (hESC) medium comprising DMEM/F12 (Gibco #12400024) with MEM NEAA (Gibco #11140050), 14.3-mM sodium bicarbonate (Sigma #S5761), 1-mM L-glutamine (Sigma #S5792), 100-μM β-mercaptoethanol (Merck #M3148), 20% of knockout serum replacement (Gibco #10828028), and 10-ng/ml bFGF (R&D systems #4114-TC) and passed using collagenase IV (Gibco #17104019). To generate floating embryoid bodies (EBs), hESC colonies were dissociated with collagenase IV and plated onto Petri dishes in hESC medium. A day after floating culture, to

obtain NPCs, the EBs were treated with LDN (Selleckchem #S2618) and SB-431542 (Cayman #CAY-13031) in DMEM/F12 + Glutamax (Gibco #10565042) plus N2 and B-27 supplements (Gibco #17502048 and #12587010). The treatment was continued for 7 days, followed by plating onto growth factor reduced matrigel (BD #354230)-coated dishes in DMEM/F12 plus N2 and B-27 supplements (N2B27 medium) LDN, SB-431542, and 1-μg/ml laminin (Gibco #23017015) to facilitate the attachment of the EBs. Within a few days, rosettes were manually collected and dissociated with Accutase (Innovative Cell Technologies #AT104) and plated onto poly-L-ornithine (Sigma #P3655)/laminin-coated dishes with NPC medium (N2B27 medium plus 20-ng/ml bFGF). To differentiate NPCs into neurons, NPCs were plated into polyornithine/laminin-coated plates in N2B27 medium in the presence of 100-U/ml penicillin-streptomycin (Gibco #14140122), 200-nM ascorbic acid (Sigma), 500-μg/ml dcAMP (Selleckchem), 20-ng/ml BDNF (Peprotech #450-02), 20-ng/ml GDNF (Peprotech #450-10), and 1-μg/ml laminin, as previously described[51]. Protocols describing the use of human ESCs were approved in accordance with the ethical requirements and regulations of the Institutional Review Board of KAIST (IRB #KH2017-109). Lentivirus containing shRNA and puromycin resistant gene was infected to NPCs with MOI of ~1. Twenty-four hours post-infection, a time period for sufficient transgene expression driven by lentiviruses[52], puromycin (Thermo Fisher #A1113803, 400 ng/ml) was added into NPC culture medium. Twenty-four hours after puromycin treatment, NPCs were differentiated into neurons.

**Image acquisition and analysis**. Fixed neurons were imaged with a 63X objective oil lens on a Zeiss LSM780 Confocal Microscope driven by ZEN software (Zeiss). Images were obtained as a 1-μm Z-stack with 0.5-μm spacing. For quantification of the spine/shaft fluorescence ratio, line plots of fluorescence intensity were generated across spine heads and the adjacent dendritic shafts using ZEN software. Fluorescence intensity at each compartment was quantified from the maximum intensity corresponding to the spine and the dendrite after background subtraction. Then, ratios were calculated as the ratio between the dendritic spine and shaft. More than ten spines were measured for each neurons and each experiment was repeated at least three times in separate neuronal cultures.

**Data acquisition, statistics, and analysis**. All experiments were carried out in a blind manner. Statistical analysis was conducted using Prism 7.0 software (GraphPad Software, Inc.). Statistical details are described in Supplementary Table 1.

**Reporting summary**. Further information on research design is available in the Nature Research Reporting Summary linked to this article.

## Data availability

All data supporting the findings of this study are provided within the paper and its Supplementary information. A source data file is provided with this paper. All additional information will be made available upon reasonable request to the authors. Source data are provided with this paper.

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

## Acknowledgements
We would like to Dr Kwang-Wook Choi in the Department of Biological Sciences at KAIST for thoughtful comments on the manuscript. This work was supported by the Ministry of Health & Welfare (HI18C1077 to J.H.), NRF Grant 2017M3C7A1079692 (to H.K.), and the Institute for Basic Science (IBS-R002-A1 to J.H. and IBS-R002-D1 to E.K.).

## Author contributions
S.-G.K. and S.L. designed experiments and analyzed data; Y.L., H.K., and S.-G.K. generated mice; S.-G.K., Y.K., J.P., and H.J. conducted mouse behavioral experiments and analysis; S.-G.K. and J.P. performed biochemical experiments and analysis; S.-G.K., Y.K., W.S., J.D.R., K.K., and S.M.U. conducted slice electrophysiology experiments and analysis; S.L. and J.R. performed cultured neuron experiments, protein–protein interaction experiments, and in vitro kinase assays; S.L. and K.K. performed virus injection experiments; D.W., S.L., C.Y., and M.L. performed imaging experiments and analysis; D.K. and S.L. performed human neuron experiments; E.Y. performed FISH experiments; H.K., J.H., W.D.H., and E.K. wrote the manuscript.

## Competing interests
The authors declare no competing interests.
