## [Peer Review File · Nature Communications]

Reviewers' Comments:

Reviewer #1:

Remarks to the Author:

In this manuscript, Kim et al introduce a new mTOR-inhibitory protein, Tanc2. Tanc2+/- mice show mTOR hyperactivity, deregulation of synaptic plasticity in the hippocampus as well as behavioral deficits in spatial learning and anxiety-related tests. Application of the mTOR inhibitor rapamycin rescues these deficits. Tanc2 mutations obtained from human patients show an inability to regulate mTOR appropriately. This study is of interest due to the prominent role of mTOR signaling in various diseases, of which the relevance to brain developmental diseases such as schizophrenia and autism is highlighted here.

Overall, the study is well done and the experimental approach, particularly the biochemical / molecular work is well designed. The strength in the molecular characterization is somewhat diminished by a lack of appeal for a wider audience that results from the absence of experiments that more directly link Tanc2 / mTOR signaling to neuropsychiatric disorders. The choice to study hippocampal synaptic plasticity as well as Morris water maze learning is quite understandable, but there is no direct connection to autism or schizophrenia. This deficit cannot be outweighed by the study of a maze / anxiety test, which does not address phenotypic alterations specific to any brain developmental disorder.

The characterization of synaptic phenotypes needs to be significantly expanded to gain impact. This includes the following aspects:

- Is there a change in basic synaptic transmission?
- Is PSD-95 expression altered in Tanc2+/- mice?
- The LTP/LTD data should be shown for all ages tested in the same figure.
- Does the rapamycin treatment rescue LTD? This latter point is important, as the LTD deregulation (Fig 1d) appears to be stronger than that of LTP (Fig. 1c).

Reviewer #2:

Remarks to the Author:

Kim et al., approached a novel question about the role of Tanc2 in brain function. In this approach, they uncovered that Tanc2 had a temporal role in mediating spatial and other behavioral paradigms that are relevant to symptoms co-diagnosed people carrying Tanc2 mutations. The big take aways from the paper are: 1) loss or knockdown of Tanc2 resulted in elevated mtor activity; 2) elevated mtor activity only occurred early; 2) the lack of a phenotypes later was due to a complimentary gene, i.e. Tanc1 being expressed in an inverse temporal pattern; 3) human Tanc mutations associated with human neuropsychiatric and neurological disorders mimicked some of the mouse mtor phenotypes uncovered; 4) some mtor pathology was uncovered, potentially revealing ways that Tanc2 could operate in mtor signaling, including through a potential direct interaction with mtor. Overall, this is an interesting paper with several advances to the field. I have a few comments and concerns that I believe could strengthen the manuscript.

Broadly, the paper is based primarily on biochemical techniques which are mostly believable but also very limited. If the authors could supplement these data with other techniques, it would greatly boost the rigor of the paper; I leave this into the hands of the authors as not every experiment is easily supplemented with other types of techniques and can be dependent on available collaborators. However, if complementary experiments could be employed to further test some claims this would add a great amount of strength to the paper.

Specific comments:

- 1) For Figure 1g, the figure legend states that virus was injected into the hippocampus between days 5-14 but the methods state all mice were injected at 6 days. Also, there is no indication how long after the hippocampus was injected did the authors collect tissue. I would also recommend adding a panel showing a representative example slice expressing GFP, this would help the reader.
- 2) Lines 131-133 are a little confusing. The authors are comparing multiple variables, i.e.

upstream mtor/downstream regulators and different ages, with neither panel showing any differences. However, they claim these results contrast with each other. Did the authors intend to compare one of their other datasets here? I had difficulty because when the P14 pups are referred to in line 131, there is not a reference to which figure it is. In addition, in line 133, did the authors intend to talk about HET mice instead of WT here?

3) In Figure 2, the schema in (a) makes it seem as if the authors waited 120 days to perform the assays but the figure legends makes it seem that this was performed at 7-8 weeks, which would be closer to 53-60 days.

4) Figure 3g: The mapping of the regions in Tanc2 that bind to mtor are a good addition to the mechanism by which Tanc2 could regulate mtor function. However, I would be cautious concluding that Tanc2 directly inhibits mtor function. I highly recommend toning this conclusion down as the data in Figure 3g are quite variable and mild.

5) Lines 382-384 (methods): The authors state that 24 hours after transducing human cells with their lentivirus they added puromycin. Lentiviruses are known to take several days after transduction to begin expressing their cargo. My worry here is that the authors may have evaluated cells based on their ability to survive puromycin for an extra day rather than being selected by puromycin for those actually transduced. Please address this?

6) In the summary, the authors state that mTORC1/2 components that stimulate mTOR kinase activity strongly affect neurodevelopment, but mTOR-inhibitory mTORC1/2 components do not, questioning the role of balanced mTOR regulation in neurodevelopment. This is a little confusing and I believe incorrect as TSC1/2 and PTEN, both inhibitory components, result in drastic neurodevelopmental phenotypes when deleted. I would highly recommend changing this statement or deleting it.

7) Just an optional recommendation, but the protein schema in Figure 1 seems out of place here, however, it is well placed in Figure 6. You could delete from Figure 1 and bring up before Figure 6?

8) In the results section, the authors state that 2-5 month old mice were used, line 85, for behavioral assessments, but Supp. Figure 1 states they are 2-4 months.

9) It is unclear to me if female mice were also subjected to the Morris water maze or other tests? Lines 89-90: The statement that female mice performed similarly to the males should be modified to reflect which behaviors were tested. This should also be addressed in the later experiments presented.

10) Example images of the purity of the neuron and glia cultures are needed. You could do something as simple as a generic neuron marker, beta III tubulin or Map2, with a nuclear marker like DAPI; would expect high rate of neuron marker with DAPI in neuron cultures and DAPI only in glia cultures.

11) Please provide immunohistochemical/immunofluorescent/native GFP expression data to show the efficiency of viral transduction in the hippocampal slices obtained from transduced brains.

Reviewer #3:

Remarks to the Author:

This manuscript by Kim and colleagues provides evidence that Tanc2 is a negative regulator of mTORC1 and mTORC2 signaling in mouse and human neurons. They demonstrate this using molecular/biochemical assays and further show the functional relevance of loss of Tanc2 using Tanc2^{+/-} mice, which have alterations in synaptic physiology and behavior. The authors further demonstrate that Tanc2's expression and function is most relevant during early postnatal development and that other mTOR regulators (Tanc1 and Depton) have more impact at later developmental stages.

The strengths of the study are that it is novel and very comprehensive, incorporating multiple types of assays in multiple systems (molecular to behavioral). The inclusion of in vitro and in vivo mouse models and human neurons with patient mutations is a strength. The paper is mechanistic and reveals potential molecular-level mechanisms for Tanc2 suppression of mTOR activity. It also has disease relevance as Tanc2 has been linked to a number of neurodevelopmental and psychiatric disorders. The developmental switch between Tanc2 and Tanc1/Depton is particularly interesting.

Overall, the authors convincingly show that Tanc2 is a novel mTOR inhibitor that has

neurodevelopmental outcomes, particularly during early postnatal periods. That said, there are some minor to moderate concerns (detailed below) that should be addressed in a revision prior to publication.

Major points:

In the abstract and at several points in the manuscripts the authors present the argument that “[stimulators of] mTOR kinase activity strongly affect neurodevelopment but that mTOR-inhibitory components do not”. This does not seem to be a very relevant argument as canonical “mTORopathies”, which are neurodevelopmental disorders, result from loss of function of mTOR negative regulators including Tuberous Sclerosis Complex (TSC1/2) and Cowden’s syndrome (PTEN). I don't think there is any question that balanced mTOR regulation is essential for neurodevelopment. Perhaps the authors can use another argument/rationale to frame their study.

The ages used differ quite a bit across experiments, i.e. behavior was done at 2-5 months, LTP was affected in 7-8 week old animals but not in 4-5 week old mice. However, basal synaptic transmission and LTD were only measured at 3-4 weeks. It would be important to know how basal synaptic properties are affected in 7-8 week old mice to know whether the reduced LTP was due to an occlusion effect (i.e. caused by already potentiated synaptic transmission). In addition, mTOR signaling is assessed at P14 in Tanc2^{+/-} mice but not at later ages, again making it a bit challenging to link up the different phenotypes. For example, is it definitely the case that the mTOR hyperactivity only occurs during early development and that this sets the stage for aberrant synaptic function and behavior later in life? Or could it be that mTOR is hyperactive early, not during adolescence, but becomes hyperactive again later in life (~7 weeks+), which directly causes the synaptic and behavior phenotypes observed in adults?

Related to this, for the AAV-Cre injection, it says this was done “(P5-14)” – does this mean that the injection was done in animals who were in the range of P5 to P14 or was it done at P5 and then the effects were analyzed at P14? Also, I’m not sure the authors can conclude that juvenile loss of Tanc2 does not cause mTOR hyperactivity as the P28 time point was assessed in Het animals (with developmental haploinsufficiency) and not with the conditional model, which would allow a time-dependent full deletion of Tanc2 at later ages that could be directly compared with the P5-14 deletion. This section may need to be revised for clarity or additional experiments may be needed to support these conclusions (top of pg. 7).

It seems important to include negative controls for the Tanc2-mTOR interaction studies in Fig. 3 – i.e. show proteins that do not bind to Tanc2 as controls (perhaps other mTORC1/2 components or signaling pathway members). Does purified Tanc2 also inhibit mTORC2 activity as suggested by the in vivo experiments?

Fig. 3G is an important graph for the paper to show that purified Tanc2 inhibits mTORC1 activity, however, the effects on p-S6K are variable and not 100% convincing. The data in HEK cells in Sup. Fig. 7 is more convincing, perhaps this could be moved to the main figure.

The human neuron experiments are lacking characterization. The authors should show example images of these cultures (preferably stained for neuronal markers) or western blots to validate that these are indeed neurons.

General comment, for some of the analyses/figures, the statistics and comparisons are not clearly stated or explained (see some specific examples below). This is important as some of the effects are somewhat subtle.

Minor points/suggestions:

It would be helpful to show the full survival curve for Tanc2^{+/-} mice. Do any mice die after P7? Do they have normal body weight?

mTOR phosphorylation at S2448 may not be a good indicator of mTOR activity. This is a minor concern as the authors show S6K, S6 and 4E-BP1, which are more reliable read-outs.

The modulation of mTORC1 by Tanc2 is an interesting part of the paper. It would be helpful to know if other mTORC2 targets besides p-Akt 473 are impacted.

It is unusual in Fig. 1 that p-S6 is not strongly altered as this is usually a very sensitive read-out for mTORC1 activity. The authors show stronger regulation in other cell types/systems, therefore this is not a major concern.

For the in vitro studies in Fig. 5, it would be helpful for the authors to clarify the timeline of the experiments in the results or main figure. i.e. Was the knock-down induced at DIV 7 and then neurons were assessed at DIV 14? (likewise for the DIV21-28 experiments). This is shown with a schematic in Sup. Fig. 11 but it could be shown in the main text.

NPC-derived human neurons are unlikely to be truly "mature" within 2 weeks.

In the discussion the authors suggest that rapamycin could be used to block the Tanc2-mTOR interaction to promote mTOR activity; however, it is well known that rapamycin is a potent mTORC1 inhibitor.

The authors nicely show that Tanc2 may be most relevant for suppressing mTOR activity in neurons compared to glia but it would be useful if they could discuss the expression patterns of Tanc2 in the brain. i.e. is it expressed pan-neuronally or does it have cell type-specific expression?

What statistics were used to compare the conditions for Fig. 2c and e. The legends state a two-way ANOVA (presumably for the escape latency graph) but this does not seem appropriate for the number of crossings and swim speed graphs (i.e. one-way ANOVA would be more relevant – please indicate which post-hoc comparisons were done, i.e. were all groups compared to each other or just to the control?)

The legends for Fig. 5 say that n of 3 independent experiments were done, however, each bar shows 4 data points.

In Fig. 6 it is unclear why shTanc2 #1 does not show a significant effect while #2 does – despite that the data for #2 are more variable. What post-hoc tests were done? (i.e. just WT vs each shRNA?)

On pg. 7 the authors reference Fig. 1f, however, I believe this should be Fig 1g.

The title for Sup Fig. 1 says "Impaired hyperactivity..." perhaps the authors just mean "Hyperactivity..."?

The time course showing Tanc1/2 expression over time in cultures is very interesting (Sup. Fig. 10). It would be interesting to look at the developmental expression of Tanc1 and Tanc2 in vivo and potential changes with age (this is just a suggestion, not a required experiment)

Point-by-point response to review comments

Re: NCOMMS-20-26654

Tanc2-dependent direct and regulated mTOR inhibition balances mTORC1/2 signaling in developing mouse and human neurons

REVIEWER COMMENTS

Reviewer #1 (Remarks to the Author):

In this manuscript, Kim et al introduce a new mTOR-inhibitory protein, Tanc2. Tanc2^{+/-} mice show mTOR hyperactivity, deregulation of synaptic plasticity in the hippocampus as well as behavioral deficits in spatial learning and anxiety-related tests. Application of the mTOR inhibitor rapamycin rescues these deficits. Tanc2 mutations obtained from human patients show an inability to regulate mTOR appropriately. This study is of interest due to the prominent role of mTOR signaling in various diseases, of which the relevance to brain developmental diseases such as schizophrenia and autism is highlighted here.

Overall, the study is well done and the experimental approach, particularly the biochemical / molecular work is well designed. The strength in the molecular characterization is somewhat diminished by a lack of appeal for a wider audience that results from the absence of experiments that more directly link Tanc2 / mTOR signaling to neuropsychiatric disorders. The choice to study hippocampal synaptic plasticity as well as Morris water maze learning is quite understandable, but there is no direct connection to autism or schizophrenia. This deficit cannot be outweighed by the study of a maze / anxiety test, which is does not address phenotypic alterations specific to any brain developmental disorder.

→ We appreciate the encouraging comments of the reviewer! As correctly pointed out by the reviewer, our results on Tanc2 and mTOR regulation are currently not directly associated with any particular neuropsychiatric disorders. However, we would like to emphasize that this is the first paper characterizing Tanc2-dependent inhibition of mTOR, and we plan to pursue disease-related mechanisms in more detail in follow-up studies.

The characterization of synaptic phenotypes needs to be significantly expanded to gain impact. This includes the following aspects:

- Is there a change in basic synaptic transmission?

→ We tested basal synaptic transmission at *Tanc2*^{+/-} SC-CA1 synapses at ~7–8 weeks and found that there is no genotype difference (**Fig. 1c**). In addition, paired-pulse facilitation was not changed at these synapses (**Fig. 1d**). These results suggest the decreased LTP at this stage does not involve changes in basal transmission or presynaptic release.

- Is PSD-95 expression altered in Tanc2^{+/-} mice?

→ We performed additional immunoblot analysis for PSD-95 and found that PSD-95 levels were unaltered at P14, P28, or P52 (**Fig. 2a,c,d**).

- The LTP/LTD data should be shown for all ages tested in the same figure.

→ We now show all the LTP/LTD data from all the ages tested in a single figure (**Fig. 1**). The mTOR hyperactivity results presented in the original **Fig. 1** were moved to **Fig. 2** (new), where they were combined with other mTOR hyperactivity data, which were originally presented in supplementary figures or newly produced by revision experiments.

- Does the rapamycin treatment rescue LTD? This latter point is important, as the LTD deregulation (Fig 1d) appears to be stronger than that of LTP (Fig. 1c).

→ We agree with the reviewer that the LTD impairment is indeed stronger than the LTP impairment and that testing if rapamycin could rescue the LTD phenotype is important. However, we have to point out that rapamycin treatment was performed during P10–35, and two more weeks (P35–49) are given to mice for the recovery from rapamycin treatment and minimization of potential side effects of the drug treatment/long handling, which makes it possible to measure electrophysiological characteristics only at ~P49. We have to point out that measuring LTD after P49 (7 weeks) is very difficult at least in our hands; we usually measure LTD during P14–21 (2–3 weeks). We clarified this in Results, and the reviewer's understanding of this practical limitation would be much appreciated.

Reviewer #2 (Remarks to the Author):

Kim et al., approached a novel question about the role of Tanc2 in brain function. In this approach, they uncovered that Tanc2 had a temporal role in mediating spatial and other behavioral paradigms that are relevant to symptoms co-diagnosed people carrying Tanc2 mutations. The big take aways from the paper are: 1) loss or knockdown of Tanc2 resulted in elevated mtor activity; 2) elevated mtor activity only occurred early; 3) the lack of a phenotypes later was due to a complimentary gene, i.e. Tanc1 being expressed in an inverse temporal pattern; 4) human Tanc mutations associated with human neuropsychiatric and neurological disorders mimicked some of the mouse mtor phenotypes uncovered; 5) some mtor pathology was uncovered, potentially revealing ways that Tanc2 could operate in mtor signaling, including through a potential direct interaction with mtor. Overall, this is an interesting paper with several advances to the field. I have a few comments and concerns that I believe could strengthen the manuscript.

Broadly, the paper is based primarily on biochemical techniques which are mostly believable but also very limited. If the authors could supplement these data with other techniques, it would greatly boost the rigor of the paper; I leave this into the hands of the authors as not every experiment is easily supplemented with other types of techniques and can be dependent on available collaborators. However, if complementary experiments could be employed to further test some claims this would add a great amount of strength to the paper.

→ We appreciate the encouraging and helpful comments of the reviewer. We fully agree with the reviewer that employment of other technologies could have helped increase the rigor of the study. In fact, the design of Fig. 4 in the original manuscript (now Fig. 6) was for FRET experiments (CFP-Tanc2 and YFP-mTOR), but forward/reverse Tanc2-mTOR interactions occurred too slowly, taking ~4 to 24 hours, and thus was not ideal for FRET experiments. We thus had to generate the data of Tanc2-mTOR colocalizations using the same constructs (Fig. 6).

Specific comments:

1) For Figure 1g, the figure legend states that virus was injected into the hippocampus between days 5-14 but the methods state all mice were injected at 6 days. Also, there is no indication how long after the hippocampus was injected did the authors collect tissue. I would also recommend adding a panel showing a representative example slice expressing GFP, this would help the reader.

→ We added a new figure panel showing the study design (virus injection during P5–14 and immunoblot analysis at P14) and GFP expression (**Fig. 2e**). The relevant Methods section was corrected. We also added a similar diagram to **Fig. 2g**.

2) Lines 131-133 are a little confusing. The authors are comparing multiple variables, i.e. upstream mtor/downstream regulators and different ages, with neither panel showing any differences. However, they claim these results contrast with each other. Did the authors intend to compare one of their other datasets here? I had difficulty because when the P14 pups are referred to in line 131, there is not a reference to which figure it is. In addition, in line 133, did the authors intend to talk about HET mice instead of WT here?

→ My apologies for the confusion! We tried to compare the results from P14 and P28. In addition, we tried to emphasize that the minimal impacts of Tanc2 haploinsufficiency at P28 is in line with the previously reported decrease in the expression of Tanc2 in WT mice (ref 14). We clarified them in the text as follows: “This contrasts with results from *Tanc2*^{+/-} pups (P14) and suggests that the function of Tanc2 is age-dependent, consistent with the strong decrease in Tanc2 protein levels in the wild-type (WT) mouse brain after P14.” was changed to “These results at P28 and P52 contrast with those from younger *Tanc2*^{+/-} mice (P14) (**Fig. 2a,b**), which suggest that Tanc2 functions are age-dependent, consistent with the decreasing Tanc2 expression in the WT mouse brain after P14”.

3) In Figure 2, the schema in (a) makes it seem as if the authors waited 120 days to perform the assays but the figure legends makes it seem that this was performed at 7-8 weeks, which would be closer to 53-60 days.

→ We added a new schematic diagram, which now better shows that the electrophysiological experiments were performed during P49–56 and the behavioral experiments were performed during P49–120 (**Fig. 3a**).

4) Figure 3g: The mapping of the regions in Tanc2 that bind to mtor are a good addition to the mechanism by which Tanc2 could regulate mtor function. However, I

would be cautious concluding that Tanc2 directly inhibits mtor function. I highly recommend toning this conclusion down as the data in Figure 3g are quite variable and mild.

→ We agree with the reviewer that we need to tone down our conclusions unless we know further details on how Tanc2 inhibits mTOR. We corrected the texts in Summary, Results, and Figure legends.

5) Lines 382-384 (methods): The authors state that 24 hours after transducing human cells with their lentivirus they added puromycin. Lentiviruses are known to take several days after transduction to begin expressing their cargo. My worry here is that the authors may have evaluated cells based on their ability to survive puromycin for an extra day rather than being selected by puromycin for those actually transduced. Please address this?

→ We appreciate the careful comment. We have to point out that a previous study has measured the intensity of fluorescent proteins driven by lentivirus and shown that sufficient transgene (puromycin and GFP) expression can occur during the first two days, including the time window of 24–48 hours (see the figure below) (*Lab Chip* **10**, 1967-1975 (2010)), during which (24–48 hours) we attempted our puromycin selection. We clarified this in Methods with the citation of this reference.

6) In the summary, the authors state that mTORC1/2 components that stimulate mTOR kinase activity strongly affect neurodevelopment, but mTOR-inhibitory mTORC1/2 components do not, questioning the role of balanced mTOR regulation in neurodevelopment. This is a little confusing and I believe incorrect as TSC1/2 and PTEN, both inhibitory components, result in drastic neurodevelopmental phenotypes when deleted. I would highly recommend changing this statement or deleting it.

→ We agree with the reviewer and changed the statement in Summary as follows: “mTORC1/2 components that stimulate mTOR kinase activity strongly affect neurodevelopment, but mTOR-inhibitory mTORC1/2 components do not, questioning the role of balanced mTOR regulation in neurodevelopment.” was changed to “However, components of the mTORC1/2 complexes that negatively regulate mTOR kinase activity are not fully understood.” We also changed related texts in Introduction and Discussion.

7) Just an optional recommendation, but the protein schema in Figure 1 seems out of place here, however, it is well placed in Figure 6. You could delete from Figure 1

and bring up before Figure 6?

→ As requested, we removed the protein schema from **Fig. 1a**. The schema is actually brought up in **Fig. 3** (now **Fig. 4**) for the first time, but not in **Fig. 6**, in the original manuscript; **Fig. 4** (current) describes the deletion variants of Tanc2 used to determine Tanc2 domains required for mTOR interaction.

8) In the results section, the authors state that 2-5 month old mice were used, line 85, for behavioral assessments, but Supp. Figure 1 states they are 2-4 months.

→ My apologies! We corrected “2–4 months” to “2–5 months” in **Supplementary Figs. 1–3** legends.

9) It is unclear to me if female mice were also subjected to the Morris water maze or other tests? Lines 89-90: The statement that female mice performed similarly to the males should be modified to reflect which behaviors were tested. This should also be addressed in the later experiments presented.

→ We modified the sentence as follows: “Female adult *Tanc2*^{+/-} mice showed behavioral abnormalities similar to those of males (**Supplementary Fig. 3**).” was changed to “Female adult *Tanc2*^{+/-} mice showed largely similar behavioral abnormalities; hyperactivity (open-field) and anxiolytic-like behavior (elevated plus-maze) but normal depression-like behavior (forced-swim and tail-suspension) (**Supplementary Fig. 3**).” We also made similar changes in the text for male behaviors.

10) Example images of the purity of the neuron and glia cultures are needed. You could do something as simple as a generic neuron marker, beta III tubulin or Map2, with a nuclear marker like DAPI; would expect high rate of neuron marker with DAPI in neuron cultures and DAPI only in glia cultures.

→ We now show that cultured neurons are positive for both DAPI and NeuN (neuronal marker) but not GFAP (astrocyte marker), while glial cells are positive for both DAPI and GFAP but not NeuN staining (**Supplementary Fig. 8c**).

11) Please provide immunohistochemical/immunofluorescent/native GFP expression data to show the efficiency of viral transduction in the hippocampal slices obtained from transduced brains.

→ We now show the image of GFP-expressing neurons in the hippocampus of virus-injected mice (**Fig. 2e,g**).

Reviewer #3 (Remarks to the Author):

This manuscript by Kim and colleagues provides evidence that Tanc2 is a negative regulator of mTORC1 and mTORC2 signaling in mouse and human neurons. They demonstrate this using molecular/biochemical assays and further show the functional relevance of loss of Tanc2 using *Tanc2*^{+/-} mice, which have alterations in synaptic

physiology and behavior. The authors further demonstrate that Tanc2's expression and function is most relevant during early postnatal development and that other mTOR regulators (Tanc1 and Deptor) have more impact at later developmental stages.

The strengths of the study are that it is novel and very comprehensive, incorporating multiple types of assays in multiple systems (molecular to behavioral). The inclusion of in vitro and in vivo mouse models and human neurons with patient mutations is a strength. The paper is mechanistic and reveals potential molecular-level mechanisms for Tanc2 suppression of mTOR activity. It also has disease relevance as Tanc2 has been linked to a number of neurodevelopmental and psychiatric disorders. The developmental switch between Tanc2 and Tanc1/Deptor is particularly interesting.

Overall, the authors convincingly show that Tanc2 is a novel mTOR inhibitor that has neurodevelopmental outcomes, particularly during early postnatal periods. That said, there are some minor to moderate concerns (detailed below) that should be addressed in a revision prior to publication.

→ We appreciate the encouraging and helpful comments of the reviewer.

Major points:

In the abstract and at several points in the manuscripts the authors present the argument that “[stimulators of] mTOR kinase activity strongly affect neurodevelopment but that mTOR-inhibitory components do not”. This does not seem to be a very relevant argument as canonical “mTORopathies”, which are neurodevelopmental disorders, result from loss of function of mTOR negative regulators including Tuberous Sclerosis Complex (TSC1/2) and Cowden’s syndrome (PTEN). I don’t think there is any question that balanced mTOR regulation is essential for neurodevelopment. Perhaps the authors can use another argument/rationale to frame their study.

→ We fully agree with the reviewer and, accordingly, changed our arguments/rationales in several different places in the manuscript, including Summary, Introduction, and Discussion (changed texts are indicated in green).

The ages used differ quite a bit across experiments, i.e. behavior was done at 2-5 months, LTP was affected in 7-8 week old animals but not in 4-5 week old mice. However, basal synaptic transmission and LTD were only measured at 3-4 weeks. It would be important to know how basal synaptic properties are affected in 7-8 week old mice to know whether the reduced LTP was due to an occlusion effect (i.e. caused by already potentiated synaptic transmission). In addition, mTOR signaling is assessed at P14 in Tanc2^{+/-} mice but not at later ages, again making it a bit challenging to link up the different phenotypes. For example, is it definitely the case that the mTOR hyperactivity only occurs during early development and that this sets the stage for aberrant synaptic function and behavior later in life? Or could it be that mTOR is hyperactive early, not during adolescence, but becomes hyperactive again later in life (~7 weeks+), which directly causes the synaptic and behavior phenotypes

observed in adults?

→ To address these comments, we first measured basal synaptic transmission and paired-pulse facilitation at *Tanc2*^{+/-} SC-CA1 synapses at 7–8 weeks, a stage relevant to adult behaviors. We found that there are no significant changes in both parameters (**Fig. 1c,d**). These results suggest that altered basal transmission or presynaptic release does not cause LTP suppression.

Second, we measured mTOR-related signals at P52, in addition to P14 and P28, and found that there is no mTOR hyperactivity (**Fig. 2d**). Therefore, mTOR hyperactivity is observed at ~P14 but not at P28 or P56, suggesting that the early mTOR hyperactivity might set the stage for the behavioral abnormalities at juvenile and adult stages.

Related to this, for the AAV-Cre injection, it says this was done “(P5-14)” – does this mean that the injection was done in animals who were in the range of P5 to P14 or was it done at P5 and then the effects were analyzed at P14? Also, I’m not sure the authors can conclude that juvenile loss of *Tanc2* does not cause mTOR hyperactivity as the P28 time point was assessed in Het animals (with developmental haploinsufficiency) and not with the conditional model, which would allow a time-dependent full deletion of *Tanc2* at later ages that could be directly compared with the P5-14 deletion. This section may need to be revised for clarity or additional experiments may be needed to support these conclusions (top of pg. 7).

→ Our apologies for the confusion. We corrected the schematic diagram, which now clearly shows that the virus injection was performed at P5 (single time point), and the immunoblot analysis was performed at P14, after 9-day expression (**Fig. 2e**).

Second, we performed the experiment injecting the Cre-expressing virus during P19–28 (in addition to P5–14) and found no mTORC1/2 hyperactivity (**Fig. 2g,h**), suggesting that juvenile loss of *Tanc2* does not cause mTOR hyperactivity.

It seems important to include negative controls for the *Tanc2*-mTOR interaction studies in Fig. 3 – i.e. show proteins that do not bind to *Tanc2* as controls (perhaps other mTORC1/2 components or signaling pathway members). Does purified *Tanc2* also inhibit mTORC2 activity as suggested by the in vivo experiments?

→ First, we attempted a negative control experiment using Deptor, a known inhibitor of mTORC1, and found that Deptor does not interact with *Tanc2* (**Fig. 4d**).

Second, we performed an in vitro kinase assay for mTORC2 where HEK cell-expressed mTOR and Rictor proteins were used to phosphorylate Akt, a substrate of mTORC2, and found that *Tanc2* inhibits mTORC2-dependent Akt (S473) phosphorylation (**Fig. 5c**), suggesting that *Tanc2* inhibits mTORC2 in addition to mTORC1.

Fig. 3G is an important graph for the paper to show that purified *Tanc2* inhibits mTORC1 activity, however, the effects on p-S6K are variable and not 100% convincing. The data in HEK cells in Sup. Fig. 7 is more convincing, perhaps this could be moved to the main figure.

→ We moved **Supplementary Fig. 7** to a main figure panel (**Fig. 5a**); previous main

Fig. 3 was divided into two main figures (**Fig. 4** [for Tanc2 interaction with mTOR] and **Fig. 5** [for Tanc2 inhibition of mTOR activity]) to make the whole figure not too crowded.

The human neuron experiments are lacking characterization. The authors should show example images of these cultures (preferably stained for neuronal markers) or western blots to validate that these are indeed neurons.

→ We now show colocalizations of human NPCs with Nestin and SOX2 (NPC markers) and human neurons with MAP2 and Tuj1 (neuronal markers) (**Supplementary Fig. 10**).

General comment, for some of the analyses/figures, the statistics and comparisons are not clearly stated or explained (see some specific examples below). This is important as some of the effects are somewhat subtle.

→ We made these aspects clearer in the revised manuscript; clarifications were made in the following figures/legends: **Fig. 3b–e**; **Fig. 5a–c**; **Fig. 6a,b,e**; **Supplementary Fig. 1d**, **Supplementary Fig. 3b**, **Supplementary Fig. 6b**. We also tried to use Bonferroni test throughout the manuscript for consistency, which did not alter the main conclusions. One exception was **Fig. 3e**, where Tukey (but not Bonferroni) test yielded a significant difference, and this was clearly stated in the figure legend.

Minor points/suggestions:

It would be helpful to show the full survival curve for *Tanc2*^{+/-} mice. Do any mice die after P7? Do they have normal body weight?

→ The survival rate of *Tanc2*^{+/-} mice at P5 is ~56% of the expected value (100%), suggesting that *Tanc2* haploinsufficiency leads to substantial lethality (embryonic or early postnatal). The survival rate at P110 was ~44% of the expected value, indicative of continuing but relatively mild lethality during adolescence and adulthood. *Tanc2*^{+/-} mice have body weights of ~90% compared with WT mice at P110. We clarified these results in the Results section.

mTOR phosphorylation at S2448 may not be a good indicator of mTOR activity. This is a minor concern as the authors show S6K, S6 and 4E-BP1, which are more reliable read-outs.

→ We agree with the reviewer that mTOR-S2448 is not a good indicator of mTOR activity, as compared with S6K, S6, and 4E-BP1. Along this line, we repeated the experiments for **Fig. 6a–f** (in the original manuscript) and found that S6K and S6 are indeed better markers than mTOR and 4E-BP1 at least under this experimental condition; *Tanc2* (WT) strongly inhibited S6K-T389 and S6-S236 phosphorylation but not mTOR-S2448 or 4E-BP1-T37/46 phosphorylation. Decreases in mTOR phosphorylation by *Tanc2* were not evident in this experiment, unlike the results in the original manuscript, likely because we used, this time, transfected and exogenous S6K (not endogenous as in the original experiments; due to weak

endogenous S6K phosphorylation in HEK cells), which might have affected mTOR phosphorylation.

Importantly, we found that the human mutations, which we showed to suppress Tanc2-dependent mTOR inhibition in the original manuscript, were no longer effective when we used S6K and S6 phosphorylation as markers (see the results below). Given that S6K and S6 are more reliable markers relative to mTOR under this experimental context, we concluded that human Tanc2 mutations do not affect Tanc2-dependent mTOR inhibition and thus decided to eliminate the results on human Tanc2 mutations from the manuscript (original **Fig. 6a–f**). This decision is to publish only most reliable data, and we hope that this could be understood. However, the remaining results in Fig. 6 show another important conclusion that Tanc2 knockdowns in human NPCs and neurons induce mTOR hyperactivity (now **Fig. 8a–c**).

The modulation of mTORC1 by Tanc2 is an interesting part of the paper. It would be helpful to know if other mTORC2 targets besides p-Akt 473 are impacted.

→ We tested PKC α , an additional marker of mTORC2 activity, and found that the levels of p-PKC α (S657), or total PKC α , were not altered in the *Tanc2*^{+/-} brain at P14, P28 or P52 (**Fig. 2a,c,d**; description added to Results). However, we have to point out that the increases in p-Akt (S473) and p-GSK3 β (S9) at P14 (**Fig. 2a**), together with our new results that Tanc2 inhibits Akt phosphorylation in vitro (**Fig. 5c**), support the conclusion that Tanc2 inhibits mTORC2 in addition to mTORC1.

It is unusual in Fig. 1 that p-S6 is not strongly altered as this is usually a very sensitive read-out for mTORC1 activity. The authors show stronger regulation in other cell types/systems, therefore this is not a major concern.

→ We agree with the reviewer, which was the reason why we commented on this in the original text.

For the in vitro studies in Fig. 5, it would be helpful for the authors to clarify the timeline of the experiments in the results or main figure. i.e. Was the knock-down induced at DIV 7 and then neurons were assessed at DIV 14? (likewise for the DIV21-28 experiments). This is shown with a schematic in Sup. Fig. 11 but it could be shown in the main text.

→ You are correct. We clarified in the revised figure that Tanc2 knockdown in cultured mouse neurons was initiated at DIV 7 and finished at DIV 14, or during DIV 21–28 (old **Fig. 5a,c**; now **Fig. 7a,c**), similar to the clarity of the schema shown in **Supplementary Fig. 11** (now **Supplementary Fig. 8**).

NPC-derived human neurons are unlikely to be truly “mature” within 2 weeks.

→ We agree and corrected the description in figure legends.

In the discussion the authors suggest that rapamycin could be used to block the Tanc2-mTOR interaction to promote mTOR activity; however, it is well known that rapamycin is a potent mTORC1 inhibitor.

→ We deleted this sentence; thanks for the careful reading!

The authors nicely show that Tanc2 may be most relevant for suppressing mTOR activity in neurons compared to glia but it would be useful if they could discuss the expression patterns of Tanc2 in the brain. i.e. is it expressed pan-neuronally or does it have cell type-specific expression?

→ We now show by FISH (fluorescent in situ hybridization) experiments that Tanc2 expression is detected in both Vglut1/2-positive excitatory neurons and Gad1/2-positive inhibitory neurons in the mouse brain at P7 and P14 (**Supplementary Fig. 9**).

What statistics were used to compare the conditions for Fig. 2c and e. The legends state a two-way ANOVA (presumably for the escape latency graph) but this does not seem appropriate for the number of crossings and swim speed graphs (i.e. one-way ANOVA would be more relevant – please indicate which post-hoc comparisons were done, i.e. were all groups compared to each other or just to the control?)

→ We used two-way ANOVA for Fig. 2c,e but now use one-way ANOVA for the number of crossings and swim speed in Fig. 2c (now **Fig. 3c**) and open/closed arm time in Fig. 2e (now **Fig. 3e**). We now indicate all the significant and non-significant changes in the graphs (**Fig. 3c–e**) and clarified the posthoc tests used in figure legends. As mentioned above, for **Fig. 3e**, Tukey (but not Bonferroni) test yielded a significant difference, and this was clearly stated in the figure legend.

The legends for Fig. 5 say that n of 3 independent experiments were done, however, each bar shows 4 data points.

→ Four independent experiments are correct; our apologies! We corrected the figure legend (now **Fig. 7**).

In Fig. 6, it is unclear why shTanc2 #1 does not show a significant effect while #2 does – despite that the data for #2 are more variable. What post-hoc tests were done? (i.e. just WT vs each shRNA?)

→ It could be the distinct properties of the two knockdown constructs such as differences in the strengths of target gene binding, time courses of target gene knockdown, or compensatory cellular responses to adjust Akt activity, although further details remain to be determined. We commented on this in the figure legend (now **Fig. 8**). We have to emphasize, however, that the overall effects of the two independent knockdown constructs are similar.

On pg. 7 the authors reference Fig. 1f, however, I believe this should be Fig 1g.

→ Thanks! We corrected it (now **Fig. 2e,f**).

The title for Sup Fig. 1 says “Impaired hyperactivity…” perhaps the authors just mean “Hyperactivity…”?

→ Thanks! We corrected it.

The time course showing Tanc1/2 expression over time in cultures is very interesting (Sup. Fig. 10). It would be interesting to look at the developmental expression of Tanc1 and Tanc2 in vivo and potential changes with age (this is just a suggestion, not a required experiment).

→ Thanks! While the panel A shows the results from cultured neurons, the panels B and C show the in vivo results from the mouse brain and the quantification. We further clarified this in the figure legend (now **Supplementary Fig. 7**).

Reviewers' Comments:

Reviewer #1:

Remarks to the Author:

I have no further comments, my previous critiques have been appropriately addressed.

Reviewer #2:

Remarks to the Author:

The authors have done a great job addressing all of my recommendations.

- Daniel Vogt

Reviewer #3:

Remarks to the Author:

The authors have made a good effort to address the majority of the reviewers' concerns in the revision and the manuscript has been improved. The conclusion that Tanc2 is a negative regulator of mTOR signaling is an important finding, as is the developmental dynamics of this regulation, which is complementary with other mTORC1 regulators. Given that TANC2 mutations are associated with neuropsychiatric disease, this paper provides important new mechanisms that have potential disease relevance.

I have just a few remaining comments, which are minor:

- 1) In the abstract, I would suggest revising the newly added sentence that "...components of the mTORC1/2 complexes that negatively regulate mTOR kinase activity are not fully understood". There is a huge body of literature studying negative regulators of mTORC1 including TSC1/2, PTEN, NF1, DEPDC5, etc. Perhaps the authors could say something along the lines of 'we still don't fully understand all of the upstream signaling components that can regulate mTOR signaling, especially in neurons'.
- 2) Related to this, in lines 59-60 I would make it clear that the authors are referring to members of the mTORC complex itself - i.e. take Deptor and PRAS40 out of parentheses - as stated previously, other negative regulators of mTORC1 have profound neurodevelopmental impacts (TSC1/2, PTEN, etc)
- 3) In line 124, the authors discuss the "activities" of different proteins but perhaps they mean 'total levels of' or 'expression of' since "activity" per se hasn't been directly measured.
- 4) The schematics showing the timelines of the experiments are very helpful and make things more clear; however, in lines 146-147, I would write 'injected at P5 and analyzed at P14' (and injected at P19 and analyzed at P28) or something similar since "P5-14" and "P19-28" are still unclear.
- 5) In lines 162-164, it is appreciated that LTD experiments at later ages may be more difficult than at younger ages, however, many labs do LTD experiments at this age so I would remove this sentence.
- 6) TANC2 should be capitalized in line 318 (human gene)

Point-by-point response to review comments

Re: NCOMMS-20-26654A

Tanc2-mediated mTOR inhibition balances mTORC1/2 signaling in the developing mouse brain and human neurons

REVIEWER COMMENTS

Reviewer #1 (Remarks to the Author):

I have no further comments, my previous critiques have been appropriately addressed.

→ We appreciate the final comments of the reviewer.

Reviewer #2 (Remarks to the Author):

The authors have done a great job addressing all of my recommendations.

→ We appreciate the final comments of the reviewer.

Reviewer #3 (Remarks to the Author):

The authors have made a good effort to address the majority of the reviewers' concerns in the revision and the manuscript has been improved. The conclusion that Tanc2 is a negative regulator of mTOR signaling is an important finding, as is the developmental dynamics of this regulation, which is complementary with other mTORC1 regulators. Given that TANC2 mutations are associated with neuropsychiatric disease, this paper provides important new mechanisms that have potential disease relevance.

→ We appreciate these summary comments.

I have just a few remaining comments, which are minor:

1) In the abstract, I would suggest revising the newly added sentence that "...components of the mTORC1/2 complexes that negatively regulate mTOR kinase activity are not fully understood". There is a huge body of literature studying negative regulators of mTORC1 including TSC1/2, PTEN, NF1, DEPDC5, etc. Perhaps the authors could say something along the lines of 'we still don't fully understand all of the upstream signaling components that can regulate mTOR signaling, especially in neurons'.

→ We incorporated the suggested point into the Abstract as follows: "However, components of the mTORC1/2 complexes that negatively regulate mTOR kinase activity are not fully understood." was changed to "However, we do not fully understand all of the upstream signaling components that can regulate mTOR

signaling, especially in neurons.”

2) Related to this, in lines 59-60 I would make it clear that the authors are referring to members of the mTORC complex itself - i.e. take Deptor and PRAS40 out of parentheses - as stated previously, other negative regulators of mTORC1 have profound neurodevelopmental impacts (TSC1/2, PTEN, etc).

→ We incorporated the reviewer’s point into this part of Introduction as follows: “However, loss of mTORC inhibitors (Deptor and PRAS40) has no significant impact on embryonic development or postnatal growth or survival^{1,2}.” was changed to “However, deletion of Deptor or PRAS40 in mice has no significant impact on embryonic development or postnatal growth or survival^{1,2}. There are upstream negative regulators of mTOR such as TSC1/2 and PTEN, NF1, and DEPDC5 that have strong impacts on neurodevelopment³⁻¹⁶, although these regulators are not fully understood.”

3) In line 124, the authors discuss the "activities" of different proteins but perhaps they mean 'total levels of' or 'expression of' since "activity" per se hasn't been directly measured.

→ To clarify this, we changed the text as follows: “Intriguingly, mTOR activity, measured by mTOR phosphorylation (S2448) in immunoblot analyses, was markedly (~5-fold) increased...” was changed to “Intriguingly, mTOR activity, indirectly measured by total levels of mTOR phosphorylation (S2448) in immunoblot analyses, was markedly (~5-fold) increased...”

4) The schematics showing the timelines of the experiments are very helpful and make things more clear; however, in lines 146-147, I would write 'injected at P5 and analyzed at P14' (and injected at P19 and analyzed at P28) or something similar since "P5-14" and "P19-28" are still unclear.

→ We made the suggested changes as follows: “Injection of AAV1-hSyn-Cre-EGFP into the hippocampus of *Tanc2*^{fl/fl} mice during P5–14, but not during P19–28, to produce local homozygous knockout of *Tanc2* induced hyperphosphorylation...” was changed to “Local homozygous knockout of *Tanc2* in the hippocampus of *Tanc2*^{fl/fl} mice by the injection of AAV1-hSyn-Cre-EGFP at P5 and analysis at P14, but not the injection at P19 and analysis at P28, induced hyperphosphorylation...”.

5) In lines 162-164, it is appreciated that LTD experiments at later ages may be more difficult than at younger ages, however, many labs do LTD experiments at this age so I would remove this sentence.

→ We removed the sentence.

6) TANC2 should be capitalized in line 318 (human gene).

→ Corrected.

We would like to sincerely appreciate the careful comments of the reviewer.

References

1. Malla, R., Wang, Y., Chan, W.K., Tiwari, A.K. & Faridi, J.S. Genetic ablation of PRAS40 improves glucose homeostasis via linking the AKT and mTOR pathways. *Biochemical pharmacology* **96**, 65-75 (2015).
2. Caron, A., *et al.* Loss of hepatic DEPTOR alters the metabolic transition to fasting. *Mol Metab* **6**, 447-458 (2017).
3. Duman, R.S. Ketamine and rapid-acting antidepressants: a new era in the battle against depression and suicide. *F1000Res* **7**(2018).
4. Borrie, S.C., Brems, H., Legius, E. & Bagni, C. Cognitive Dysfunctions in Intellectual Disabilities: The Contributions of the Ras-MAPK and PI3K-AKT-mTOR Pathways. *Annu Rev Genomics Hum Genet* **18**, 115-142 (2017).
5. Lipton, J.O. & Sahin, M. The neurology of mTOR. *Neuron* **84**, 275-291 (2014).
6. Jaworski, J. & Sheng, M. The growing role of mTOR in neuronal development and plasticity. *Molecular neurobiology* **34**, 205-219 (2006).
7. Switon, K., Kotulska, K., Janusz-Kaminska, A., Zmorzynska, J. & Jaworski, J. Molecular neurobiology of mTOR. *Neuroscience* **341**, 112-153 (2017).
8. Costa-Mattoli, M. & Monteggia, L.M. mTOR complexes in neurodevelopmental and neuropsychiatric disorders. *Nature neuroscience* **16**, 1537-1543 (2013).
9. Hoeffler, C.A. & Klann, E. mTOR signaling: at the crossroads of plasticity, memory and disease. *Trends in neurosciences* **33**, 67-75 (2010).
10. Ehninger, D. & Silva, A.J. Rapamycin for treating Tuberous sclerosis and Autism spectrum disorders. *Trends in molecular medicine* **17**, 78-87 (2011).
11. Toldo, I., *et al.* Tuberous sclerosis-associated neuropsychiatric disorders: a paediatric cohort study. *Developmental medicine and child neurology* **61**, 168-173 (2019).
12. Iffland, P.H., 2nd, Carson, V., Bordey, A. & Crino, P.B. GATORopathies: The role of amino acid regulatory gene mutations in epilepsy and cortical malformations. *Epilepsia* **60**, 2163-2173 (2019).
13. Parenti, I., Rabaneda, L.G., Schoen, H. & Novarino, G. Neurodevelopmental Disorders: From Genetics to Functional Pathways. *Trends in neurosciences* **43**, 608-621 (2020).
14. Curatolo, P., Moavero, R. & de Vries, P.J. Neurological and neuropsychiatric aspects of tuberous sclerosis complex. *The Lancet. Neurology* **14**, 733-745 (2015).
15. Rademacher, S. & Eickholt, B.J. PTEN in Autism and Neurodevelopmental Disorders. *Cold Spring Harbor perspectives in medicine* **9**(2019).
16. Shilyansky, C., Lee, Y.S. & Silva, A.J. Molecular and cellular mechanisms of learning disabilities: a focus on NF1. *Annual review of neuroscience* **33**, 221-243 (2010).